# CRITICAL INFLUENCE OF OVERPARAMETERIZATION ON SHARPNESS-AWARE MINIMIZATION

## ABSTRACT

Training overparameterized neural networks often yields solutions with varying generalization capabilities, even when achieving similar training losses. Recent evidence indicates a strong correlation between the sharpness of a minimum and its generalization error, leading to increased interest in optimization methods that explicitly seek flatter minima for improved generalization. Despite its contemporary relevance to overparameterization, however, this sharpness-aware minimization (SAM) strategy has not been studied much yet as to exactly how it is affected by overparameterization. In this work, we analyze SAM under varying degrees of overparameterization, presenting both empirical and theoretical findings that reveal its critical influence on SAM's effectiveness. First, we conduct extensive numerical experiments across diverse domains, demonstrating that SAM consistently benefits from overparameterization. Next, we attribute this phenomenon to the interplay between the enlarged solution space and increased implicit bias resulting from overparameterization. Furthermore, we show that this effect is particularly pronounced in practical settings involving label noise and sparsity, and yet, sufficient regularization is necessary. Last but not least, we provide other theoretical insights into how overparameterization helps SAM achieve minima with more uniform Hessian moments compared to SGD, and much faster convergence at a linear rate.

## 1 INTRODUCTION

The remarkable success of deep learning can largely be attributed to the increasing size of neural networks. As these networks grow deeper and wider, they have demonstrated exceptional performance across a wide range of applications (Kaplan et al., 2020; Alayrac et al., 2022; Dehghani et al., 2023; Radford et al., 2023). This raises an intriguing question: why do such large—and thus potentially overparameterized—neural networks work so well? Although a complete understanding remains elusive, research suggests that overparameterization can positively impact various aspects of the learning process, including even generalization (Neyshabur et al., 2017; Du & Lee, 2018). In particular, overparameterized neural networks often exhibit convexity-like behavior during optimization, making all local minima globally optimal, and thereby, allowing global minima to be found with local optimization methods such as gradient descent (Kawaguchi, 2016; Du et al., 2019).

However, not all global minima are necessarily equivalent; converging to different minima can yield a large disparity of generalization capabilities, despite their same level of training loss reaching almost zero (Keskar et al., 2017; Gunasekar et al., 2018). One plausible explanation for such an implicit phenomenon is that generalization is somewhat negatively correlated with the sharpness of the loss landscape, *i.e.*, flat minima tend to generalize better than sharp ones (Chaudhari et al., 2017; Jiang et al., 2020). This calls for new ways to guide the optimization process to converge to flat minima, and various strategies have been suggested to this end (Izmailov et al., 2018; Foret et al., 2021; Orvieto et al., 2022; Zhao et al., 2022). Indeed, it has been observed in many studies that reducing sharpness can enhance generalization performance (Bahri et al., 2022; Chen et al., 2022b; Qu et al., 2022).

Despite the sharpness minimization scheme being developed in the context of overparameterization, the precise impact of overparameterization on this scheme has not been studied much in the literature. In this work, we systematically investigate the effects of overparameterization on sharpness-aware minimization (SAM) (Foret et al., 2021). Specifically, we conduct extensive experiments to precisely measure the impact of overparameterization across a diverse set of tasks, ranging from standard tasks in computer vision and natural language processing, to molecular property prediction, and

further, to video game in reinforcement learning. To gain further insight into the results, we perform detailed investigations into the interactions between overparameterization and SAM through visual inspection of the solution space on a simple regression setting as well as analyzing the influence of overparameterization on the implicit bias of SAM. Furthermore, we study how overparameterization influences SAM under various conditions, including label noise, sparsity, and regularization. Last but not least, we explore other implications of overparameterization on SAM through theoretical analyses, including the characteristics of the attained minima and the convergence rate.

Our key contributions and findings are summarized as follows.

- Section 3.    We perform extensive experiments across eight workloads of datasets and models at varying scales, spanning synthetic, vision, language, chemistry, and game domains. We observe that overparameterization consistently improves the generalization benefit of SAM[1]. This phenomenon is general and previously unknown[2].

- Section 4.    We propose hypotheses to understand this general phenomenon, positing that two factors may be at play: (i) overparameterization first increases the number of simpler and flatter solution candidates, and (ii) it also increases the implicit bias of SAM. These are verified with standard experiments in both synthetic and realistic settings.

- Section 5.    We present the merits and caveats of overparameterization in employing SAM in practice: (i) the benefit of overparameterization for SAM is more pronounced under label noise and sparsity, while (ii) sufficient regularization is needed. This can serve as a useful guidance for practitioners.

- Section 6.    We develop theoretical analyses[3] on linear stability and convergence: under overparameterization, (i) linearly stable minima for SAM are flatter and have more uniformly distributed Hessian moments compared to SGD, and (ii) a stochastic SAM can converge at a linear rate. These are also numerically verified.

- Overall.    We discover that overparameterization has *critical* influences on SAM. Both empirical performance and theoretical aspects of SAM all improve with overparameterization. In other words, SAM may not take its advantage over SGD without overparameterization.

## 2    BACKGROUND

Let us consider the general unconstrained optimization problem:

$$\min_x f(x) \tag{1}$$

where $f : \mathbb{R}^d \to \mathbb{R}$ is the objective function to minimize, and $x \in \mathbb{R}^d$ is the optimization variable. Based on recent studies that indicate a strong correlation between the sharpness of $f$ at a minimum and its generalization error (Keskar et al., 2017; Dziugaite & Roy, 2017; Jiang et al., 2020), Foret et al. (2021) suggest to turn (1) into a min-max problem of the following form

$$\min_x \max_{\|\epsilon\|_2 \leq \rho} f(x + \epsilon) \tag{2}$$

where $\epsilon$ and $\rho$ denote some perturbation added to $x$ and its norm bound, respectively. Thus, the goal is now to seek $x$ that minimizes $f$ in its $\epsilon$-neighborhood, such that the objective landscape becomes locally flat. Taking the first-order Talyor approximation of $f$ at $x$ and solving for optimal $\epsilon^\star$ gives the following update rule for SAM:

$$x_{t+1} = x_t - \eta \nabla f \left( x_t + \rho \frac{\nabla f(x_t)}{\|\nabla f(x_t)\|_2} \right). \tag{3}$$

SAM has been shown to be effective for improving generalization performance compared against SGD (Chen et al., 2022b; Kaddour et al., 2022; Bahri et al., 2022), and subsequent works have analyzed

---

[1]By "generalization benefit", we mean the improvement made by SAM over SGD in validation accuracy.

[2]While evidence of the similar observation can be found in the literature (Chen et al., 2022b), no prior work has conducted experiments or confirmed this phenomenon at any scale comparable to ours.

[3]We note that these are not intended to directly support Section 3 and 4, which we discuss in Section 7.

various aspects of SAM under different settings including its convergence rates (Andriushchenko & Flammarion, 2022; Mi et al., 2022; Si & Yun, 2023) and implicit bias (Compagnoni et al., 2023; Wen et al., 2023; Andriushchenko et al., 2023).

Meanwhile, a considerable amount of evidence has indicated the benefit of overparameterization for training neural networks. Besides the empirical success witnessed across different domains (Kaplan et al., 2020; Radford et al., 2021; Dehghani et al., 2023), overparameterization turns all local minima into global ones in theory enabling local methods to succeed under non-convex settings (Kawaguchi, 2016; Du et al., 2019). Researchers have also proved the power of overparameterization to enable much faster convergence (Ma et al., 2018; Vaswani et al., 2019; Meng et al., 2020) and better generalization (Allen-Zhu et al., 2019; Brutzkus & Globerson, 2019). To our knowledge, however, previous work has mostly focused on non-sharpness-aware optimizers, and the effects of overparameterization on SAM has been left rather unattended despite its contemporary significance to large-scale training trends and widespread usage in practice.

## 3 KEY OBSERVATION: SAM IMPROVES WITH OVERPARAMETERIZATION

| Workload # | Domain | Task | Dataset | Architecture | Model |
|---|---|---|---|---|---|
| 1 | Synthetic | Regression | Synthetic | MLP | Two-layer MLP |
| 2 | Vision | Image classification | MNIST | MLP | LeNet-300-100 |
| 3 | Vision | Image classification | CIFAR-10 | CNN | ResNet-18 |
| 4 | Vision | Image classification | ImageNet | CNN | ResNet-50 |
| 5 | Language | PoS tagging | Universal Dependencies | Transformer | Encoder-only Transformer |
| 6 | Language | Sentiment classification | SST-2 | RNN | LSTM |
| 7 | Chemistry | Graph property prediction | ogbg-molpcba | GNN | GCN |
| 8 | Game | Proximal policy optimization | Atari Breakout | CNN | Five-layer CNN |

Table 1: Summary of evaluation workloads. They cover eight different datasets spanning five domains and six tasks at varying scale, and include eight neural network models of five different architecture types. For each workload, we test up to ten different models of varying degrees of parameterization.

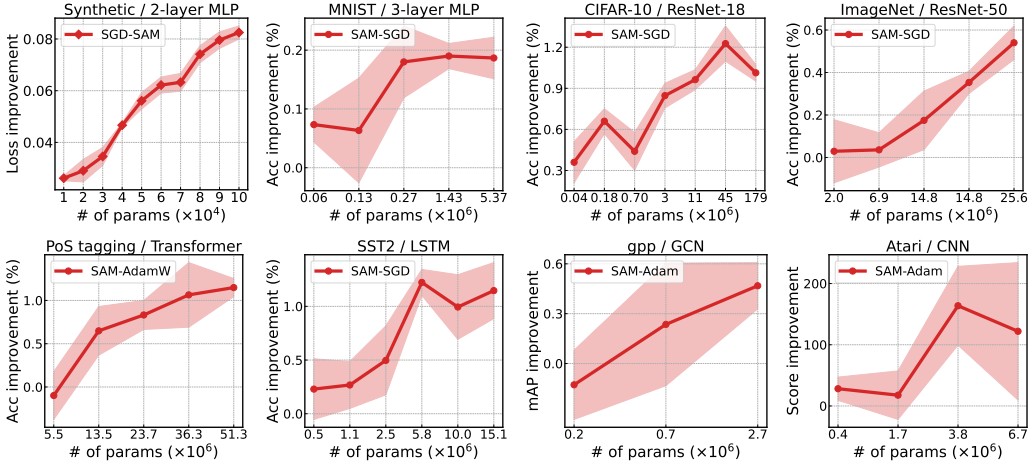

Figure 1: Improvement in validation metrics by SAM. The generalization benefit of SAM tends to increase as the model becomes more overparameterized. We present the full results including the absolute metrics for SAM and baseline optimizers in Figure 7 of Appendix B.

SAM is introduced to find flat minima and thereby improve generalization performance in practice. In this work, we are interested in whether and how this improvement is affected by overparameterization. In order to understand any potential relationship between SAM and overparameterization, we first focus on precisely measuring the effect of overparameterization. More specifically, we conduct a wide range of deep learning experiments (see Table 1 for the summary of all tested workloads), and observe how the generalization improvement made by SAM changes as with more parameters.

As a result, we find a strong and consistent trend that SAM improves with overparameterization in all tested cases (see Figure 1). To elaborate, initially, SAM does *not* work much better than the non-sharpness-aware baseline optimizer (*i.e.*, SGD or Adam family depending on the default choice) when the model is at relatively low number of parameters; it only starts to improve with more parameters and makes a clear distinction at very large number of parameters. We emphasize that this holds true for a wide variety of architectures (MLP, CNN, RNN, GCN, Transformer) and datasets of different domains (Synthetic, Vision, Language, Chemistry, Game) under a rigorous hyperparameter search (see Appendix A.1 for the full experiment details).

This result possibly indicates that SAM is more effective, when (and possibly only when) applied to overparameterized models. On the other hand, the increased generalization performance of SAM with more parameters renders a promising avenue, given that the modern neural network models are often heavily overparameterized (Zhang et al., 2022; Dehghani et al., 2023). We note that some evidence of the similar positive influence of overparameterization for SAM can be derived in the literature (Chen et al., 2022b), however, no prior work has conducted experiments or confirmed this phenomenon at any scale comparable to ours.[4]

## 4 UNDERSTANDING WHY SAM IMPROVES WITH OVERPARAMETERIZATION

Then why does overparameterization particularly favor SAM over non-sharpness-aware optimizers? We address this question in this section to better understand the effect of overparameterization on SAM. Precisely, we posit that it is potentially due to the complementarity between overparameterization enlarging the solution space and the implicit bias of SAM driving toward flat minima; *i.e.*, once there are more diverse solutions available (including both sharp and flat minima) by overparameterization, optimizers intrinsically biased toward flat solutions (such as SAM) will more likely find such solutions than unbiased optimizers (such as GD). We support this reasonable hypothesis by demonstrating the followings: (i) SAM finds simpler and flatter solutions than GD with the enlarged solution space (Section 4.1), and (ii) the implicit bias of SAM becomes stronger with overparameterization (Section 4.2); both of these take place only when the model is overparameterized.

### 4.1 ENLARGED SOLUTION SPACE ALLOWS SAM TO FIND SIMPLER AND FLATTER SOLUTIONS

To corroborate our hypothesis, we start with a simple experiment where we train one-hidden-layer ReLU networks using full-batch SAM and GD following Andriushchenko & Flammarion (2022); we use 5, 10, 100, and 1000 hidden neurons for underparameterized to highly overparameterized cases; we run three random seeds and compare solutions obtained by SAM and GD in Figure 2.

First, we find that the solutions found by SAM are not differentiated much from those of GD when the model has no more than 10 neurons. Looking closely into the case of 10 neurons, they all seem to be roughly 4 to 6 degrees of piecewise linear functions, *i.e.*, the number of line segments for each solution is less than 10, which is the maximum possible joints that this model can have in theory. On the other hand, in the case of 100 to 1000 neurons, one can easily see that the solutions found by SAM are much simpler (and thus more likely to generalize) compared to those by GD.

Next, we also track the optimization trajectories of both SAM and GD. The trajectories are plotted along PCA directions calculated from the converged minima following Li et al. (2018). The results are illustrated in Section 4.1. We find that both SAM and GD reach solutions in a similar basin when the model is under/moderately parameterized, whereas in the overparameterized case, they reach different solutions, *i.e.*, SAM reaches a flatter solution, even though they all start from the same initial point.

These results support the idea that SAM has some implicit bias that drives itself toward a certain type of solutions (*e.g.*, simple and flat) as previously shown in prior work (Andriushchenko & Flammarion, 2022; Compagnoni et al., 2023; Wen et al., 2023). More importantly, however, these results newly reveal that *overparameterization is a critical factor in facilitating this implicit behavior of SAM*; without it the space of potential solutions decreases, and SAM may not take effect.

---

[4]As an additional result, we provide a theoretical analysis of the effect of overparameterization decreasing the test error of SAM in Appendices K to L. Precisely, however, this result only mean for SAM and is not to be confused with the relative improvement against SGD as shown in Section 3.

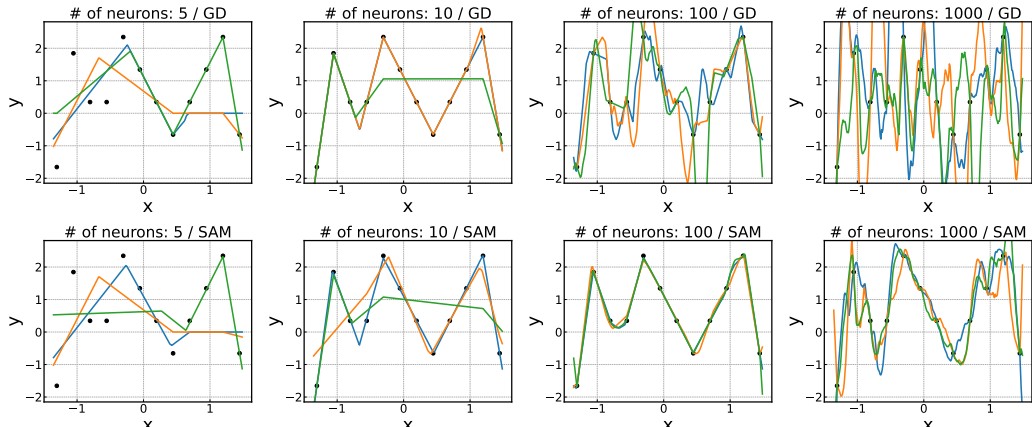

Figure 2: Solutions found by GD (top) and SAM (bottom). Both optimizers find similar solutions for under/moderately-parameterized models, whereas the solutions found by SAM are much simpler with less variance compared to those by GD for overparameterized models. Here, different colors correspond to different random seeds. See Appendix A.2 for the experiment details.

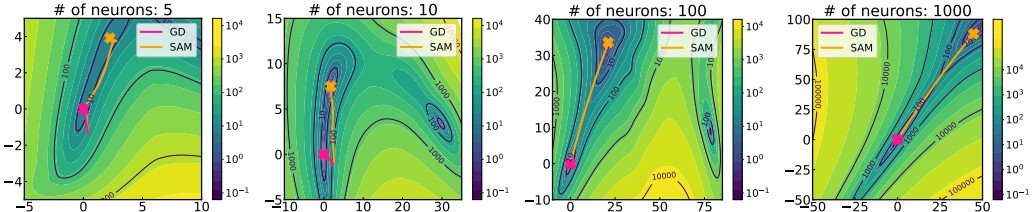

Figure 3: Optimization trajectories of GD and SAM starting from the same initial point. GD and SAM reach solutions in a similar basin for under/moderately-parameterized models, whereas they reach different solutions for overparameterized models, *i.e.*, flatter region for SAM.

### 4.2 IMPLICIT BIAS OF SAM INCREASES WITH OVERPARAMETERIZATION

While overparameterization can secure favorable conditions for SAM, it is not to be confused with guaranteeing the implicit bias of SAM taking effect. In fact, we can further relate the implicit bias of SAM to the perturbation bound $\rho$ to bridge this gap. Specifically, SAM can be proved to be an SDE model of SGD on an implicitly regularized loss (Compagnoni et al., 2023):

$$\tilde{f}(x) := f(x) + \rho \mathbb{E} \|\nabla f_\gamma(x)\|_2 \tag{4}$$

where $\gamma$ refers to some stochasticity. This indicates that SAM becomes more regularized (*i.e.*, the implicit bias is amplified) when $\rho$ increases.[5]

Our interest thus lies in seeing whether overparameterization has any effect on increasing $\rho$. Since if that is the case, it indeed means that overparameterization puts more regularization on SAM. We verify this by finding the empirically optimal perturbation bound $\rho^\star$ that yields the best generalization performance as we change the degree of overparameterization. Specifically, we take a standard deep learning task and perform an extensive grid search to find $\rho^\star$. The result is displayed in Figure 4.

Indeed, it is observed that $\rho^\star$ tends to increase as the number of parameters increases; *i.e.*, seeing from left to right, $\rho$ value that yields highest accuracy (marked as green star ⋆) tends to increase. We confirm that this trend is consistently observed for various other workloads (See Figures 10 to 13 of Appendix E for more results). This result is certainly encouraging since it supports that *the generalization benefit of SAM via implicit regularization can indeed increase by overparameterization*. Additionally, we provide a conceptual account of why intuitively it makes sense to see increasing $\rho$ with overparameterization in Appendix D.

---

[5]This holds as long as $\rho$ is not too large, by which it might overshadow minimizing $f$ and implicitly bias the optimizer toward stationary points such as saddles and maxima. Note it reduces to standard SGD when $\rho = 0$.

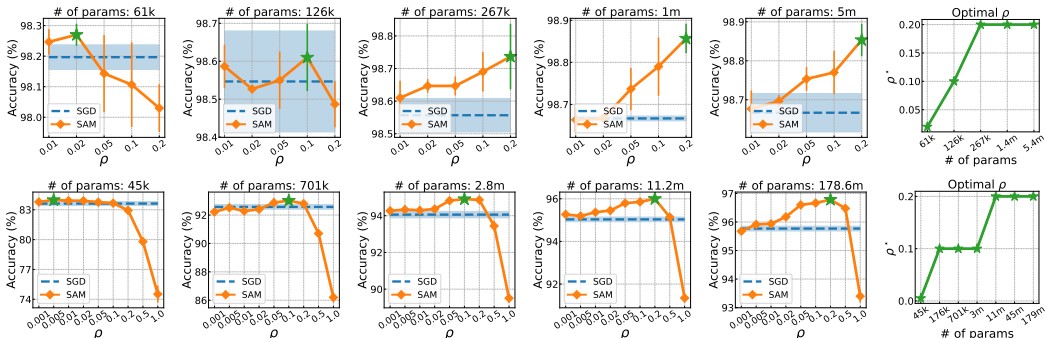

Figure 4: Validation accuracy versus $\rho$ for 3-layer MLP trained on MNIST (top) and ResNet-18 trained on CIFAR-10 (bottom). $\rho^\star$ is located to be higher with more parameters.

## 5 FURTHER MERITS AND CAVEATS OF OVERPARAMETERIZATION

In this section, we present further merits and some caveats of overparameterization. Specifically, we show that the overparameterization benefit of SAM continues to exist and becomes more evident under label noise or sparsity. We also discover that sufficient regularization is required to attain the benefit. These results could serve as a guidance to employ SAM in practice.

**Overparameterization secures the robustness of SAM to label noise**    In practice, deep learning models are often trained on noisy data (Song et al., 2022). To examine whether the overparameterization benefit for SAM continues to exist in this scenario, we introduce some label noise to training data (Angluin & Laird, 1988; Natarajan et al., 2013) and see how SAM responds. The results are reported in Figure 5a. Overall, we find SAM benefits from overparameterization significantly more than SGD in the presence of label noise. Precisely, the accuracy improvement made by SAM keeps on increasing as the model has more parameters, whereas the improvement over SGD is marginal for less parameterized models. Notably, this trend is more pronounced with a higher noise level; *e.g.*, it rises from $5\%$ to nearly $50\%$ at the highest noise rate. Notably, it is previously known that SAM is robust to label noise compared to SGD (Foret et al., 2021; Baek et al., 2024), and yet, this result newly reveals that overparameterization plays a profound role in securing the robustness of SAM.

**SAM benefits from sparse overparameterization.**    There has been a recent interest in employing sparsity to train large models to alleviate the computation and memory costs (Hoefler et al., 2021; Mishra et al., 2021). To test the effect of overparameterization on SAM under this setting, we introduce a varying degree of sparsity to an overparameterized model at initialization (Lee et al., 2019) such that the number of parameters matches the original dense model. The results are reported in Figure 5b. We observe that the generalization improvement tends to increase as the model becomes more sparsely overparameterized; more precisely, the average accuracy improvement increases from $0.4\%$ in the small dense model to around $0.8\%$ in the large sparse model. This result suggests that one can consider taking sparsification more actively when employing SAM.

**Sufficient regularization is needed to secure the benefit of overparameterization.**    We also investigate whether the overparameterization benefit for SAM continues to exist when models are prone to overfitting due to insufficient regularization (Ying, 2019). Specifically, we evaluate three cases: (a) without weight decay, (b) without early stopping, and (c) without sufficient inductive bias.[6] The results are reported in Figures 5c to 5e. We observe that the generalization improvement does not increase by simply adding more parameters. The results indicate that some level of regularization is required in practice to attain the overparameterization benefit for SAM.

---

[6]For (c), we train vision transformers that is not pre-trained on a massive dataset, which is known to lack inductive biases inherent to CNNs and thus more prone to overfitting (Lee et al., 2021; Chen et al., 2022a).

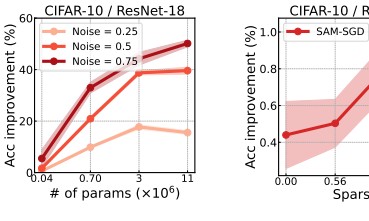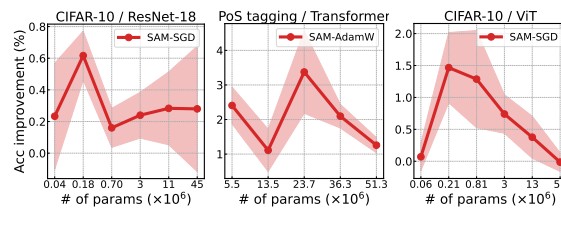

|(a) Label noise|(b) Sparsity|(c) w/o weight decay (d) w/o early stop. (e) w/o induc. bias|

Figure 5: Effect of (a) label noise, (b) sparsity, and (c-e) regularization on SAM. (a) The benefit of SAM is more pronounced with a higher noise level. (b) The improvement by SAM tends to increase in large sparse models compared to their small dense counterparts. (c-e) SAM does not always benefit from overparameterization without sufficient regularization. See Figures 14 to 18 in Appendix F for more results.

## 6 OTHER EFFECTS OF OVERPARAMETERIZATION: THEORETICAL ASPECTS

Thus far, we focused on empirically exploring how increasing number of parameters influences SAM, and discovered critical improvements in its generalization benefits. However, existing theoretical analyses on overparameterization also hint at other types of positive influences on different aspects of SAM such as convergence (Ma et al., 2018; Vaswani et al., 2019) and implicit bias (Neyshabur, 2017; Zhang et al., 2017). Despite this, we find that there is little work on explicitly verifying whether these influences extend to SAM, however.

To fill this gap, we develop theoretical analyses of the effect of overparameterization on SAM[7] in this section. Specifically, we show that (i) linearly stable minima for SAM have more uniform Hessian moments compared to SGD (Section 6.1), and (ii) SAM can converge much faster (Section 6.2), all when the model is overparameterized.

To this end, we adopt the following *interpolation* assumption to theoretically characterize overparameterization:

**Definition 6.1.** (Interpolation) There exists $x^\star$ s.t. $f_i(x^\star) = 0$ and $\nabla f_i(x^\star) = 0$ for $i = 1, \dots, n$,

To this end, which is a widely accepted notion in the literature (Ma et al., 2018; Vaswani et al., 2019). Crucially, this implies that there exists a fixed point $x^\star$ for stochastic gradient-based optimizers, which comes as an important property in the following two sections.

We also use an unnormalized version of SAM:

$$x_{t+1} = x_t - \eta \nabla f \left( x_t + \rho \nabla f(x_t) \right), \tag{5}$$

a variant empirically similar to normalized SAM which is often adopted to simplify the proof (Andriushchenko & Flammarion, 2022; Compagnoni et al., 2023).

We leave a clear note here that the aim of these analyses is to complement, rather than directly support Sections 3 and 4, by outlining theoretically guaranteed benefits of overparameterization on SAM. We discuss more about the limitations later in Section 7.

### 6.1 LINEARLY STABLE MINIMA OF SAM HAVE A MORE UNIFORM HESSIAN THAN SGD

It has been observed in Woodworth et al. (2020); Xie et al. (2021) that SGD converges to certain types of minima among many others in an overparameterized regime. We analyze the minima attained by SAM and how they compare to the minima attained by SGD from the perspective of linear stability (Wu et al., 2018; 2022), which is defined as follows:

**Definition 6.2.** (Linear stability) Consider an iterative first-order optimizer $x_{t+1} = x_t - \eta_t G(x_t)$ where $\eta_t$ denotes a step size and $G$ refers to a stochastic gradient estimate measured at the current

---

[7]We use an unnormalized version of SAM to ease the proof, a variant empirically similar to normalized SAM (Andriushchenko & Flammarion, 2022; Compagnoni et al., 2023), although recent studies have called this into question (Dai et al., 2023; Si & Yun, 2023).

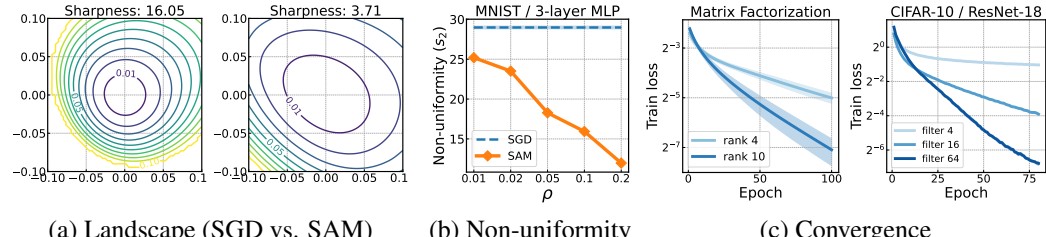

(a) Landscape (SGD vs. SAM)    (b) Non-uniformity    (c) Convergence

Figure 6: (a) Loss landscapes of SGD (left) and SAM (right) along with the corresponding sharpness $a = \lambda_{max}(H)$. SAM converges to flatter minima with lower sharpness compared to SGD. (b) Non-uniformity of Hessian for SGD and SAM. SAM has a more uniform Hessian distribution than SGD. (c) Convergence properties of SAM. As model becomes overparameterized, SAM converges much faster and closer to a linear rate. See Appendix A.3 for the experiment details.

iterate $x_t$. A minimizer $x^\star$ is called linearly stable if there exists a constant $C$ such that

$$\mathbb{E}[\|\tilde{x}_t - x^\star\|^2] \leq C\|\tilde{x}_0 - x^\star\|^2$$

for all $t > 0$ under $\tilde{x}_{t+1} = \tilde{x}_t - \nabla G(x^\star)(\tilde{x}_t - x^\star)$, *i.e.*, if it does not deviate far from $x^\star$ once arrived near a fixed point.

Here the existence of the fixed point is implied by the interpolation assumption in Definition 6.1, whereas the absence of this assumption would mean no minimum could exhibit linear stability.

With this, we provide the following theorem of a linearly stable minima for a stochastic SAM.

**Theorem 6.3.** *Let us assume $x^\star = 0$ without loss of generality. Then $x^\star$ is linearly stable for a stochastic SAM if the following is satisfied:*

$$
\begin{aligned}
\lambda_{\max} \big( (I - \eta H - \eta\rho H^2)^2 + \eta(\eta - 2\rho)(M_2 - H^2) \\
+ 2\eta^2\rho(M_3 - H^3) + \eta^2\rho^2(M_4 - H^4) \big) \leq 1
\end{aligned}
\tag{6}
$$

*where $H = \frac{1}{n}\sum_{i=1}^n H_i$ and $M_k = \frac{1}{n}\sum_{i=1}^n H_i^k$ are the average Hessian and the $k$-th moment of the Hessian at $x^\star$ over $n$ training data. Subsequently as a necessary condition of (6) it follows that*

$$0 \leq a(1 + \rho a) \leq \frac{2}{\eta}, \quad 0 \leq s_2^2 \leq \frac{1}{\eta(\eta - 2\rho)}, \quad 0 \leq s_3^3 \leq \frac{1}{2\eta^2\rho}, \quad 0 \leq s_4^4 \leq \frac{1}{\eta^2\rho^2}, \tag{7}$$

*where $a = \lambda_{max}(H), s_k = \lambda_{max}((M_k - H^k)^{1/k})$ are the sharpness and the non-uniformity of the Hessian measured with the $k$-th moment, respectively.*

The detailed proof of the theorem is provided in Appendix I.

Our result (7) suggests that SAM requires less sharp minima and more uniformly distributed Hessian moments to achieve linear stability (provided that $\rho > 0$) compared to those of SGD (Wu et al., 2018), *i.e.*, when $\rho \to 0$ in (7). While a similar result is shared by a concurrent work of Behdin et al. (2023), we further ensure that higher-order terms of Hessian moments are bounded, and interestingly, it becomes tighter for a larger $\rho$. To corroborate our result, we measure the empirical sharpness and non-uniformity of Hessian. The results are reported in Figures 6a and 6b.

## 6.2 STOCHASTIC SAM CONVERGES MUCH FASTER WITH OVERPARAMETERIZATION

Prior works have revealed the power of overparameterization for stochastic optimization methods to accelerate convergence (Ma et al., 2018; Vaswani et al., 2019; Meng et al., 2020). We prove that this benefit also extends to a stochastic SAM.

Besides the interpolation assumption we defined earlier in Definition 6.1, let us start by providing some assumptions used below.

**Definition 6.4.** (Smoothness) $f$ is $\beta$-smooth if there exists $\beta > 0$ s.t. $\|\nabla f(x) - \nabla f(y)\| \leq \beta\|x - y\|$ for all $x, y \in \mathbb{R}^d$.

**Definition 6.5.** (Polyak-Lojasiewicz) $f$ is $\alpha$-PL if there exists $\alpha > 0$ s.t. $\|\nabla f(x)\|^2 \geq \alpha(f(x) - f(x^\star))$ for all $x \in \mathbb{R}^d$.

The smoothness and the Polyak-Lojasiewicz (PL) assumptions are standard and used frequently in optimization (Gower et al., 2020; Meng et al., 2020; Nutini et al., 2022; Karimi et al., 2016). The smoothness assumption is satisfied for any neural network with smooth activation and loss function with bounded inputs (Andriushchenko & Flammarion, 2022), and the PL condition is argued to be satisfied when the model is overparameterized (Belkin, 2021; Liu et al., 2022), which we empirically verify in Figure 22b of Appendix H.

Under these assumptions, we present the following convergence theorem of a stochastic SAM:

**Theorem 6.6.** *Suppose each $f_i$ is $\beta$-smooth, $f$ is $\lambda$-smooth and $\alpha$-PL, and interpolation holds. For any $\rho \leq \frac{1}{(\beta/\alpha+1/2)\beta}$, a stochastic SAM that runs for $t$ iterations with constant step size $\eta^\star \stackrel{def}{=} \frac{\alpha-(\beta+\alpha/2)\beta\rho}{2\lambda\beta(\beta\rho+1)^2}$ gives the following convergence guarantee:*

$$\mathbb{E}_{x_t}\left[f(x_t)\right] \leq \left(1 - \frac{\alpha - (\beta + \alpha/2)\beta\rho}{2}\,\eta^\star\right)^t f(x_0).$$

We provide the full proof in Appendix J, which also contains result for the more general case of a mini-batch SAM.

This result shows that with overparameterization, a stochastic SAM can converge as fast as the deterministic gradient method at a linear convergence rate, which is much faster than the well-known sublinear rate of $\mathcal{O}(1/t)$ for SAM (Andriushchenko & Flammarion, 2022). Also, our analysis suggests that convergence is guaranteed without the bounded variance assumption and diminishing step size under overparameterization, while without overparameterization, convergence does not hold (Andriushchenko & Flammarion, 2022). This suggests that overparameterization can significantly ease the convergence of SAM. We corroborate our result empirically as well, by measuring how training proceeds with overparameterization in realistic settings. The results are plotted in Figure 6c.

# 7 CONCLUSION

In this work, we have disclosed the *critical influence of overparameterization on SAM* from empirical and theoretical perspectives. We started with an extensive evaluation to display a highly consistent trend that the generalization benefit of SAM increases with overparameterization, without which SAM may not take effect (Section 3). This led us to come up with a reasonable hypothesis to explain the benefit in terms of increased solution space and implicit bias (Section 4). In addition, we presented further merits and caveats of overparameterization in practice (Section 5). Finally, we developed theoretical advantages of overparameterization for SAM on linear stability, convergence, and generalization (Section 6). We believe these findings can bridge between overparameterization and SAM, which has been rather unattended in the literature as of yet. Nevertheless, we discuss limitations, ideas for potential future work as well as practical implications of our results below.

**Theoretical account of Section 3** The consistent trend observed in Section 3 certainly hints at the presence of a fundamental process underneath, and yet, our study does not offer a precise theory to support this phenomenon. This is largely because modeling the generalization of SAM under varying degrees of overparameterization challenges the boundaries of existing theoretical frameworks currently available in the literature. Nevertheless, drawing upon recent advancements in understanding overparameterization and generalization, we have developed plausible hypotheses to directly address this phenomenon (Section 4). We also employed rigorous theoretical frameworks to examine the effects of overparameterization on various other aspects of SAM, reinforcing the general trend of overparameterization benefits (Section 6). We believe these efforts offer valuable insights and preliminary foundations that could be instrumental in achieving a comprehensive theoretical account of Section 3 in the future.

**Other sharpness minimization schemes** Our theoretical results in Section 6 are based on an unnormalized version of SAM. This is largely driven by two reasons: (i) it appears to render minimal practical difference from the original SAM, and more crucially, (ii) it simplifies analyses as widely adopted in initial studies (Andriushchenko & Flammarion, 2022; Compagnoni et al., 2023). However, more recently, works such as Dai et al. (2023); Si & Yun (2023) have highlighted the theoretical significance of the normalization step. We plan to extend our analysis to better reflect the effect of

normalization in future work. Additionally, given that different sharpness minimization schemes can make a difference in the found minima and resulting performance (Kaddour et al., 2022; Dauphin et al., 2024), extension of our analyses to other non-SAM sharpness minimization schemes (Izmailov et al., 2018; Orvieto et al., 2022) and studying how they compare to SAM under overparameterization would be a promising avenue for future work. Nonetheless, we consider these results an initial exploration of the impact of overparameterization on SAM, setting the stage for future research.

**More ablation study**  In addition to our explanation of how overparameterization improves SAM in Section 4, we have conducted an additional ablation study to investigate the influence of other factors on the increased benefit of SAM in Appendix G. In Appendix G.1, we present preliminary investigations on the effect of increasing the depth instead of the width, where we find that the benefit of overparameterization can differ from architectures. In Appendix G.2, we test whether stronger weight decay can remove the gap between SGD and SAM, which shows that even after given much larger values of weight decay, SGD isn't able to outperform SAM on any model size. In Appendix G.3, inspired by recent studies suggesting that overparameterized models can behave like linearized models (Jacot et al., 2018) while such implicit linearization phenomenon can coincide independently of overparameterization (Chizat et al., 2019), we have tested if the increased benefit of SAM is due to linearization. As a result, we have observed that SAM underperforms SGD in the linearized regimes by more than $10\%$. This indicates that linearization is not the main factor for the increased benefit of SAM and again verifies that overparameterization itself is likely to be the main factor of the benefit.

**Potential to modern deep learning**  Our key observations in Section 3 indicate a great potential to use SAM in the modern landscape of large-scale training (Kaplan et al., 2020; Belkin, 2021). Also, our results in Section 5 further highlight its potential in the current trend where foundation models are often trained with noisy data (Radford et al., 2021; Schuhmann et al., 2022) or to employ sparsity (Frantar et al., 2024; Jiang et al., 2024). In this regard, we can possibly anticipate that the overparameterization benefit might hold even when training billion-scale foundation models (Zhang et al., 2022; Dehghani et al., 2023), which we leave to explore as future work. It would also be interesting to study how popular settings for training foundation models other than label noise or sparsity affect the benefit, such as quantization (Gholami et al., 2022), differential privacy (Yu et al., 2022), or human alignment (Ouyang et al., 2022).

## REPRODUCIBILITY

We have made a concerted effort to ensure reproducibility by following best practices and providing detailed documentation of all stages of our experimental pipeline. This includes, but is not limited to, detailed information about the experimental setup and configurations including random seeds used for all training and evaluation in Appendix A.1, the use of publicly available datasets with links and details for obtaining and preprocessing them, to name a few. Additionally, we provide our complete source code with the necessary dependencies, algorithms, and hyperparameters for reproducing the results presented in the paper as supplementary material.

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

# A   EXPERIMENTAL DETAILS

## A.1   EXPERIMENTS FOR SECTION 3

| Workload | Epochs/steps | Learning rate / decay | Weight decay | Batch size | $\rho$ search | Baseline optimizer |
|---|---|---|---|---|---|---|
| Synthetic | 100 epochs | 0.1 / step | 0.0 | 128 | $\left\{\begin{smallmatrix}0.001,0.01,0.05,0.07,0.1,\\0.2,0.3,0.5,0.7,1.0,2.0\end{smallmatrix}\right\}$ | SGD |
| MNIST/MLP | 100 epochs | 0.1 / step | 0.0001 | 128 | $\{0.01,0.02,0.05,0.1,0.2\}$ | SGD with momentum 0.9 |
| CIFAR-10/ResNet-18 | 200 epochs | 0.1 / step | 0.0005 | 128 | $\left\{\begin{smallmatrix}0.001,0.005,0.01,0.02,\\0.05,0.1,0.2,0.5,1.0\end{smallmatrix}\right\}$ | SGD with momentum 0.9 |
| ImageNet/ResNet-50 | 90 epochs | 0.1 / cosine | 0.0001 | 512 | $\{0.01,0.02,0.05,0.1,0.2\}$ | SGD with momentum 0.9 |
| PoS tagging | 75000 steps | 0.05 / inverse sqrt | 0.1 | 64 | $\{0.01,0.02,0.05,0.1,0.2,0.3,0.5\}$ | AdamW ($\beta_1 = 0.9, \beta_2 = 0.98$) |
| Sentiment classification | 30 epochs | 0.1 / constant | $3e\text{-}6$ | 64 | $\{0.01,0.02,0.05,0.1,0.2,0.3,0.5\}$ | SGD with momentum 0.8 |
| Graph property prediction | $10^5$ steps | 0.001 / constant | 0.0 | 256 | $\{0.01,0.02,0.05,0.1,0.2\}$ | Adam ($\beta_1 = 0.9, \beta_2 = 0.999$) |
| Atari game | $10^7$ steps | $2.5e\text{-}4$ / linear | 0.0 | 256 | $\{0.01,0.02,0.05,0.1,0.2\}$ | Adam ($\beta_1 = 0.9, \beta_2 = 0.999$) |
| CIFAR-10/ViT | 200 epochs | 0.1 / cosine | 0.0001 | 128 | $\{0.01,0.02,0.05,0.1,0.2\}$ | SGD with momentum 0.9 |

Table 2: Hyperparameters for each workload.

| Workload | Scaling factor | Values |
|---|---|---|
| Synthetic | # of neurons | $\{k * 100 \mid 1 \le k \le 10\}$ |
| MNIST/MLP | # of neurons | $\{[300 * p, 100 * p] \mid p \in \{0.25, 0.5, 1, 4, 10\}\}$ |
| CIFAR-10/ResNet-18 | # of convolutional filters | $\{2^k \mid 2 \le k \le 8\}$ |
| ImageNet/ResNet-50 | # of convolutional filters | $\{16 * k \mid 1 \le k \le 5\}$ |
| PoS tagging | dimension of hidden states | $\{128 * k \mid 1 \le k \le 5\}$ |
| Sentiment classification | dimension of hidden states | $\{2^k \mid 5 \le k \le 9\}$ |
| Graph property prediction | # of neurons | $\{2^k \mid 7 \le k \le 9\}$ |
| Atari game | # of convolutional filters | $\{16 * k \mid 1 \le k \le 4\}$ |
| CIFAR-10/ViT | dimension of hidden states | $\{2^k \mid 5 \le k \le 10\}$ |

Table 3: Model scaling factors and values for each workload.

For all the experiments in Section 3, we run the experiments with the same configurations over three different random seeds. Most of the experiments are conducted with a single RTX3090 GPU having 24GB memory while some experiments requiring larger memory are conducted with multiple RTX3090 GPUs. The code is implemented with JAX (Bradbury et al., 2018) and provided as a supplementary material. Many of our experiments and the hyperparameter values are based on examples provided by Flax (Heek et al., 2023) official repository.[8] The hyperparameter values and how the models are scaled for each workload are summarized in Table 2 and Table 3 respectively. We present the additional details below.

**Synthetic Regression / 2-layer MLP**   We follow the student-teacher setting from Advani et al. (2020) where the teacher is a randomly initialized 2-layer ReLU network with 200 neurons and the student is a 2-layer ReLU network with a different number of neurons. Each element for the input $x \in \mathbb{R}^{100}$ is sampled from a standard normal distribution while the target $y \in \mathbb{R}$ is calculated as the output of the teacher network added by Gaussian noise sampled from a standard normal distribution. The models are trained on 20400 training data, which is roughly the same as the number of parameters in the teacher model, and tested on the 5100 data, which is a quarter of the number of the training data.

**MNIST / 3-layer MLP**   We train LeNet-300-100 (LeCun et al., 1998) for the MNIST (LeCun et al., 2010). The learning rate decays by 0.1 after 50% and 75% of the total epochs. We scale the models while preserving the relative proportions of the number of neurons in each layer as $3 : 1$.

**CIFAR-10 / ResNet-18**   We train ResNet-18 (He et al., 2016) for the CIFAR-10 (Krizhevsky et al., 2009). We choose the hyperparameters as similar to Andriushchenko & Flammarion (2022). The learning rate decays by 0.1 after 50% and 75% of the total epochs.

**ImageNet / ResNet-50**   We train ResNet-50 (He et al., 2016) for the ImageNet (Deng et al., 2009). We choose the hyperparameters as similar to Du et al. (2022) and use a linear warmup of 5000 steps. We additionally experiment with $\rho = 0.005$ for the two smallest models.

---

[8] https://github.com/google/flax/tree/main/examples

**PoS tagging/ Transformer** We train Encoder-only Transformer (Vaswani et al., 2017) for the Universal Dependencies (Nivre et al., 2016) – Ancient Greek. We use a linear warmup of 8000 steps. We evaluate the validation accuracy once every 1000 step and report the best value except for the experiment in Figure 5d. The dimension of MLP and the number of attention heads are scaled as $4\times$ and $1/64\times$ of the dimension of the hidden states following the Flax example.

**SST / LSTM** We train LSTM (Hochreiter & Schmidhuber, 1997) for SST2 (Socher et al., 2013) where the task is a binary classification (positive/negative) of the movie reviews. We evaluate the validation accuracy for every epoch and report the best value. The embedding size is scaled as $300/256\times$ of the dimension of hidden states following the Flax example.

**Graph property prediction / GCN** We train 2-layer Graph Convolutional Networks (Kipf & Welling, 2017) for the ogbg-molpcba (Hu et al., 2020). Here, the input is a graph of a molecule where nodes and edges each represent atoms and chemical bonds. The task is a binary classification of whether a molecule inhibits HIV replication or not.

**Atari game / CNN** We train 5-layer CNNs for the Atari Breakout-v5 game (Mnih et al., 2013). We train the Actor-Critic networks (Konda & Tsitsiklis, 1999) with proximal policy optimization (Schulman et al., 2017). We evaluate the validation score once every 100 step and report the best value. We also use gradient clipping of $0.5$ for all models.

**CIFAR-10 / ViT** For the experiment in Figure 5e, we train 6-layer Vision Transformers (Dosovitskiy et al., 2021) for the CIFAR-10 (Krizhevsky et al., 2009) using the patch size of $4 \times 4$. We scale the dimension of MLP and the number of attention heads as $2\times$ and $1/32\times$ of the dimension of hidden states.

## A.2 EXPERIMENTS FOR SECTION 4

For the experiments in Figure 2 and Section 4.1, we follow the setting in Andriushchenko & Flammarion (2022).[9] Specifically, we train one-hidden-layer ReLU networks where each data has input $x \in \mathbb{R}$ and target $y \in \mathbb{R}$. Here, the networks are trained on the quadratic loss with full-batch GD or SAM with $\rho = 0.3$. Additionally, the optimization trajectories in Section 4.1 are plotted following Li et al. (2018).[10] Specifically, the trajectories are plotted along the PCA directions calculated from converged minima of two different paths from SGD and one path from SAM.

## A.3 EXPERIMENTS FOR SECTION 6

**Linear stability** For the experiments of Figures 6a and 6b, we follow the setting in Wu et al. (2018). Specifically, we set up 3-layer MLP having $[3000, 1000]$ hidden neurons with squared loss, so that the local quadratic approximation becomes precise, and train the networks on MNIST. We use 1000 random samples to calculate the non-uniformity, and all models are trained to reach near zero loss. The networks are trained with a constant learning rate of $0.1$ without weight decay or momentum.

**Convergence – Matrix Factorization** For the matrix factorization experiment in Figure 6c, we solve the following non-convex regression problem: $\min_{W_1, W_2} \mathbb{E}_{x \sim \mathcal{N}(0, I)} \|W_2 W_1 x - Ax\|^2$ where the objective function is smooth and satisfies the PL-condition (Loizou et al., 2021). We choose $A \in \mathbb{R}^{10 \times 6}$ and generate 1000 training samples, which are used for training a rank $k$ linear network with two matrix factors $W_1 \in \mathbb{R}^{k \times 6}$ and $W_2 \in \mathbb{R}^{10 \times k}$. Here, interpolation is satisfied when rank $k = 10$. We train two linear networks with $k \in \{4, 10\}$ for 100 epochs with a constant learning rate of $0.0005$ and compare the convergence speed.

---

[9] https://github.com/tml-epfl/understanding-sam/tree/main/one_layer_relu_nets

[10] https://github.com/tomgoldstein/loss-landscape

# B ABSOLUTE VALIDATION METRIC FOR SECTION 3

We present the full results of Figure 1, including the absolute validation metrics of SAM and SGD in Figure 7. There is a consistent trend that SAM improves with overparameterization in all tested cases.

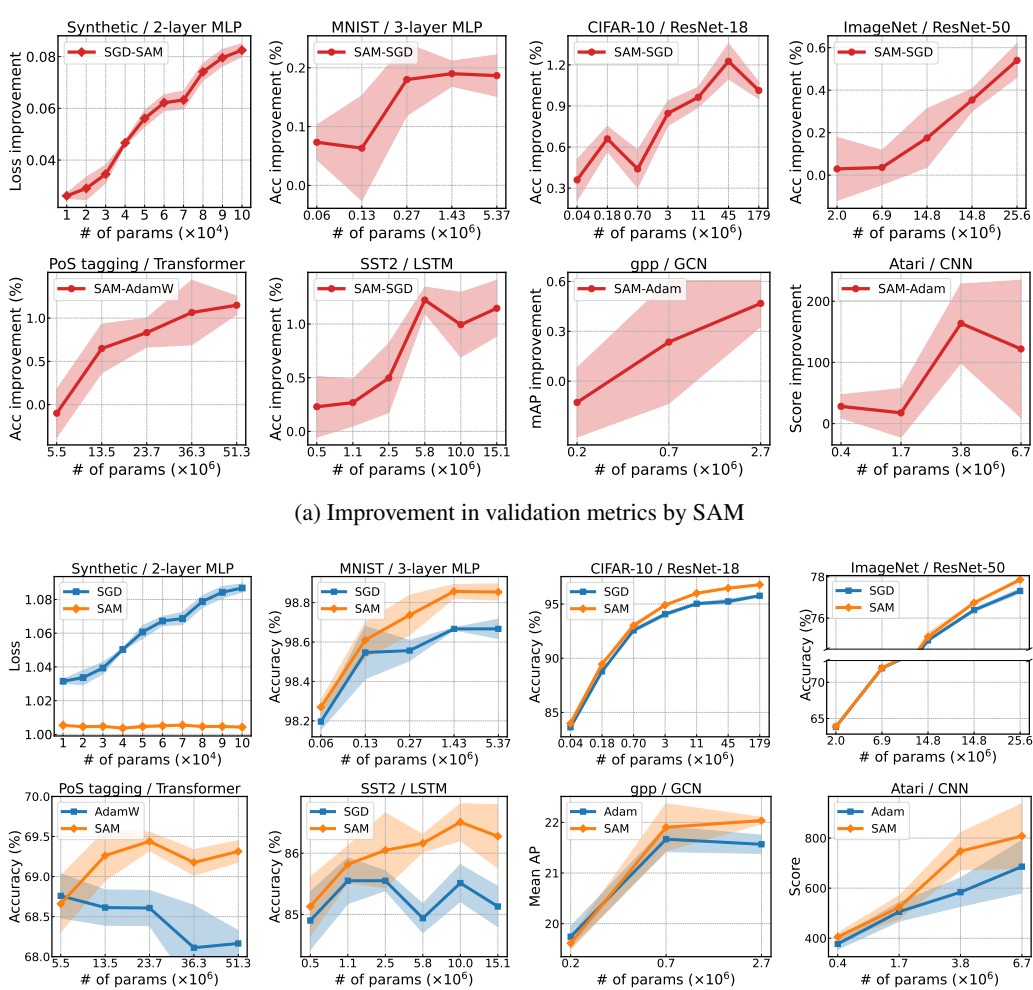

(a) Improvement in validation metrics by SAM

(b) Absolute validation metrics of SAM and baseline optimizers

Figure 7: Effects of overparameterization on SAM. Improvement in validation metrics by SAM. (a) Improvement in validation metrics by SAM and (b) the absolute metrics for SAM and baseline optimizers. The generalization benefit of SAM tends to increase as the model becomes more overparameterized.

# C MINI-BATCH TRAINING FOR SECTION 4.1

We compare the solutions of SGD and stochastic-SAM ($\rho = 0.2$) as done in Section 4.1. Here, for every training iteration, we randomly choose 6 data points for computing the loss. We use 5, 10, 100, and 1000 hidden neurons for underparameterized to highly overparameterized cases; we run three random seeds and compare solutions obtained by SAM and SGD in Figures 8. We also track the optimization trajectories of both SAM and SGD, which are plotted along PCA directions calculated from the converged minima following Li et al. (2018), which are provided in 9. We find that when the model is more overparameterized, solutions found by SAM are much more simpler and flatter compared to ones found by SGD.

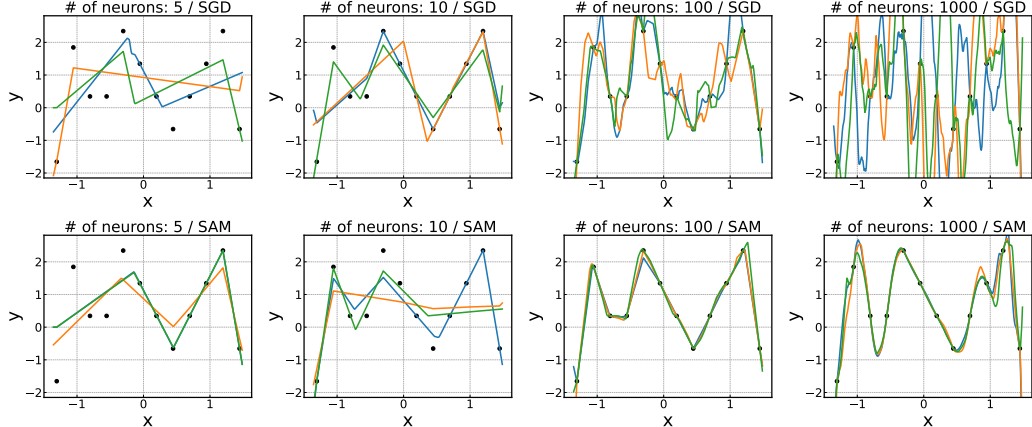

Figure 8: Solutions found by SGD (top) and stochastic-SAM (bottom). Both optimizers find similar solutions for under/moderately-parameterized models, whereas the solutions found by SAM are much simpler with less variance compared to those by SGD for overparameterized models. Here, different colors correspond to different random seeds.

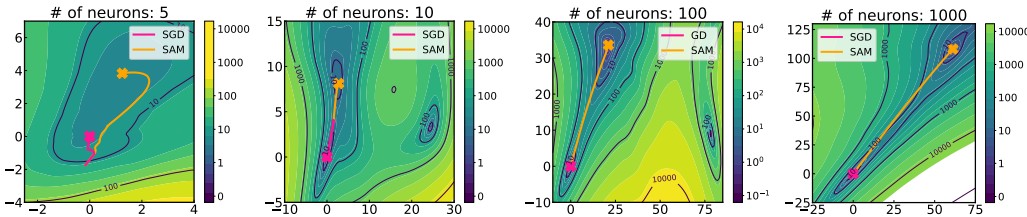

Figure 9: Optimization trajectories of SGD and stochastic-SAM starting from the same initial point. SGD and SAM reach solutions in a similar basin for under/moderately-parameterized models, whereas they reach different solutions for overparameterized models, *i.e.*, flatter region for SAM.

## D  OVERPARAMETERIZATION AND $\rho$

We can develop a simple account of why $\rho$ may need to increase with overparameterization. Firstly, the scale of perturbation $\epsilon$ controlled by $\rho$ needs to increase relatively proportionally with respect to the increase in model dimensionality in order to preserve a similar level of perturbation effect; an extreme case is when the model grows infinitely large, $\rho$ should grow together, otherwise no perturbation can be made, *i.e.*, $\|\epsilon\| \to \infty$ as $d \to \infty$.

A non-asymptotic interpretation can also be developed. Precisely, the Lipschitz bound on gradients, *i.e.*, $\left\| \nabla f \left( x + \rho \frac{\nabla f(x)}{\|\nabla f(x)\|_2} \right) - \nabla f(x) \right\|_2 \leq \beta \left\| x + \rho \frac{\nabla f(x)}{\|\nabla f(x)\|_2} - x \right\|_2 = \beta \rho$, indicates that the SAM gradient becomes more similar to the original gradient as the model gets smoother (*i.e.*, smaller smoothness constant $\beta$) with increasing size, requiring larger perturbation bound to achieve similar levels of perturbation effect. These hold under the assumption that overparameterization makes the model smoother, which we empirically confirm in Figure 22a.

## E  FULL RESULTS ON OPTIMAL PERTURBATION BOUND

Extending from Section 4, we plot the validation accuracy of SAM versus different values of $\rho$, along with their optimal value of $\rho$ for 3-layer-MLP/MNIST, ResNet-50/ImageNet, ResNet-18/CIFAR-10, and LSTM/SST2 in Figures 10 to 13, respectively. It is observed that $\rho^\star$ tends to increase as the model becomes more overparameterized; on CIFAR-10 with ResNet18, the smallest model has $\rho^\star = 0.01$ while the largest three have $\rho^\star = 0.2$.

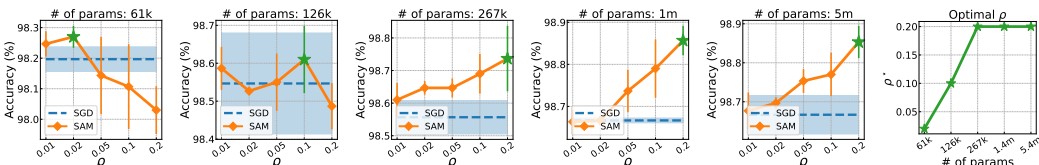

Figure 10: Validation accuracy versus $\rho$ for MNIST and 3-layer MLP.

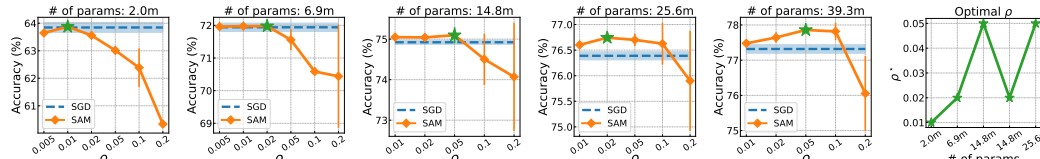

Figure 11: Validation accuracy versus $\rho$ for ResNet-50 and ImageNet.

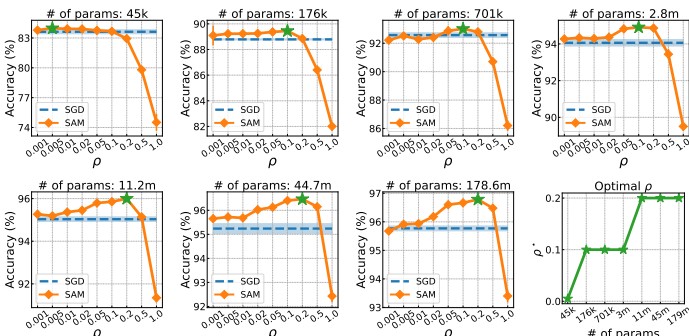

Figure 12: Validation accuracy versus $\rho$ for ResNet-18 and CIFAR-10.

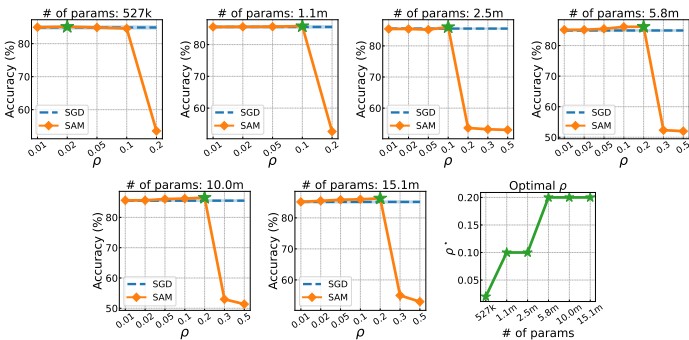

Figure 13: Validation accuracy versus $\rho$ for LSTM and SST2.

# F  ADDITIONAL RESULTS FOR SECTION 5

## F.1  LABEL NOISE

More results on the effect of overparameterization on SAM under label noise are presented in Figure 14. Overall, we find SAM benefits from overparameterization significantly more than SGD in the presence of label noise. Precisely, the accuracy improvement made by SAM keeps on increasing as the model becomes more overparameterized, and this trend is more pronounced with higher noise levels; e.g., it rises from 5% to nearly 50% at the highest noise rate.

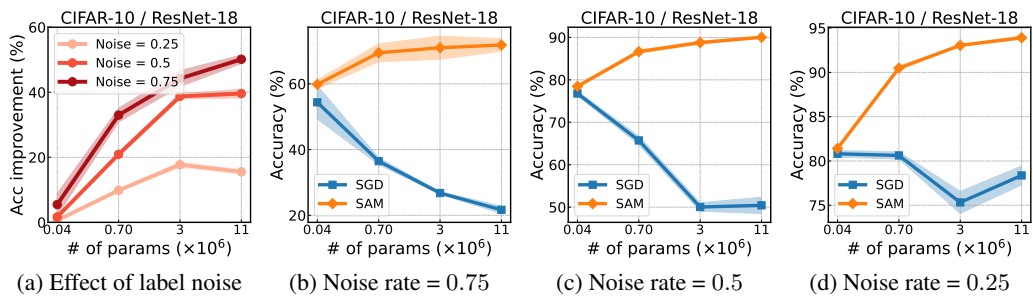

Figure 14: Effect of overparameterization on SAM under label noise for CIFAR-10 and ResNet-18. (a) SAM benefits a lot more from overparameterization than SGD; it is more pronounced with high noise level. (b-d) Under label noise, SGD tends to overfit as with more parameters unlike SAM.

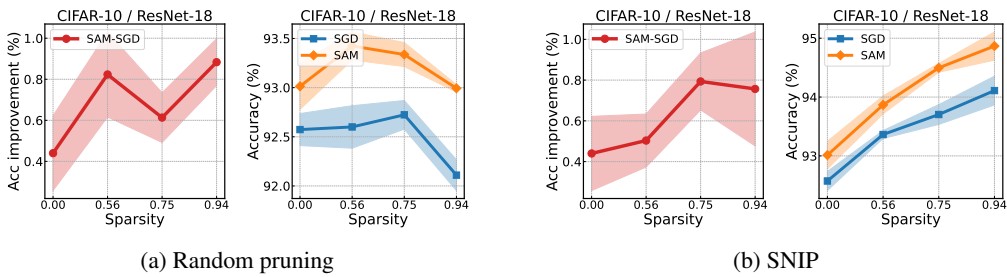

Figure 15: Effect of sparsification on SAM for ResNet-18 and CIFAR-10. Here, all the models have approximately 701k parameters. The improvement tends to increase in large sparse models compared to their small dense counterparts.

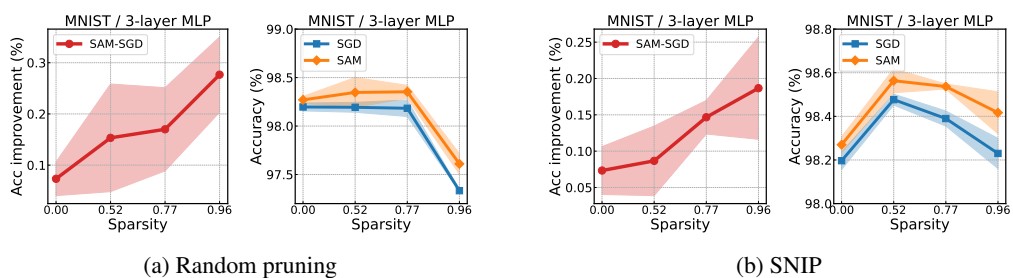

Figure 16: Effect of sparsification on SAM for MNIST and 3-layer MLP. Here, all the models have approximately 61k parameters. The improvement tends to increase in large sparse models compared to their small dense counterparts.

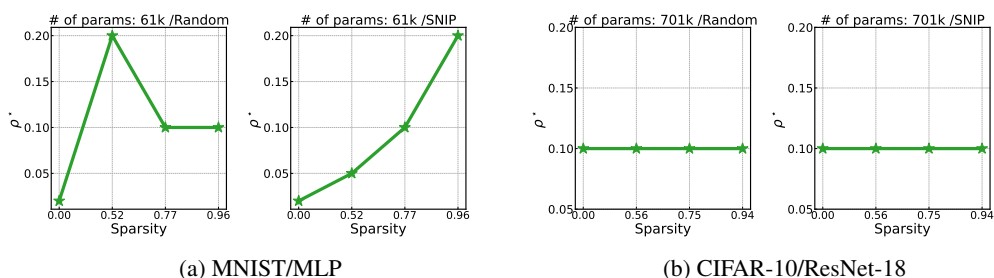

Figure 17: Effect of sparsification on $\rho^\star$. $\rho^\star$ can be sometimes different across different sparsity patterns despite having a similar number of parameters.

## F.2 SPARSE OVERPARAMETERIZATION

Additional results on the effect of sparsification on the generalization benefit of SAM are plotted in Figures 15 and 16. Here, we try two sparsification methods that do not require pertaining, random pruning and SNIP (Lee et al., 2019). For both methods, we note that the generalization improvement by SAM tends to increase as the model becomes more sparsely overparameterized.

We also plot the effect of sparsification on $\rho^\star$ in Figure 17. We find that $\rho^\star$ is sometimes different between small dense and large sparse models despite having a similar number of parameters; for the MLP of 61k parameters on MNIST, $\rho^\star$ changes over different sparsity levels and sparsification methods, but this does not generalize to the CIFAR-10 and ResNet-18. This indicates that it is not just the parameter count that affects the behavior of SAM, but some other factors such as the pattern of parameterization also have an influence on how SAM shapes training.

## F.3 REGULARIZATION

More results on the effect of regularization on SAM are presented in Figure 18. We find that overparameterization does not increase the generalization benefit of SAM. We suspect this is because the models are prone to overfitting in these cases and overparameterizing models may decrease the overall performance both for SGD and SAM; for example in Figure 18c, the validation accuracy drops after 11.2m parameters.

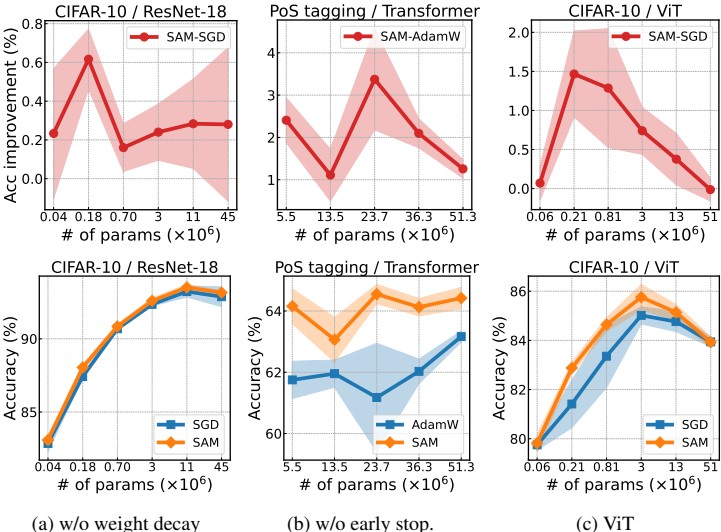

(a) w/o weight decay      (b) w/o early stop.      (c) ViT

Figure 18: Effect of overparameterization on SAM without regularization: (a) CIFAR-10/ResNet-18 without weight decay, (b) Transformer/PoS tagging without early stopping, and (c) ViT/CIFAR-10. SAM does not always benefit from overparameterization in these cases.

# G ABLATION

## G.1 EFFECT OF DEPTH

We experiment with changing the number of layers for MNIST/MLP and Cifar-10/ResNet-18. Precisely, we change the number of width-1000 hidden layers in MLP and resblock in each stage of ResNet-18 for MNIST and CIFAR-10 respectively. The results are provided in Figure 19. We find that SAM also improves with overparameterization for these cases while the increase is not significant for ResNet.

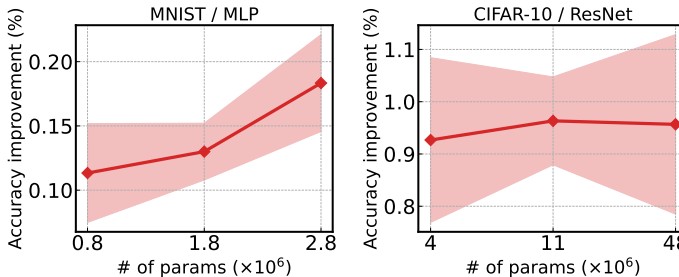

Figure 19: Improvement in validation metrics by SAM over different model depths. Deeper models tend to yield higher validation accuracy improvements. Here we change the number of width-1000 hidden layers in MLP and resblock in each stage of ResNet-18 for MNIST and CIFAR-10 respectively. Benefits of SAM also improves with overparameterization in terms of depth, although the increase is not significant for ResNet.

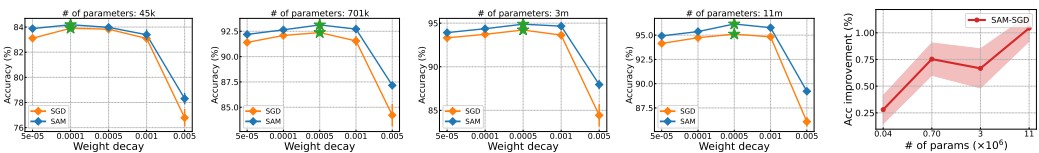

Figure 20: Effect of weight decay on validation accuracy of ResNet-18 trained on CIFAR-10 with SAM and SGD over various model scales and the improvement in validation metrics by SAM when considering weight decay. Even after being given much larger values of weight decay, SGD isn't able to outperform SAM on any model size.

## G.2 SAM VS. WEIGHT DECAY

We conduct experiments on Cifar-10/ResNet-18 for four different model sizes and five values of weight decay. The results are provided in Figure 20. We find that SGD with stronger weight decay does not compete to replace SAM for overparameterized models; for overparameterized models, using larger weight decay rather degrades the performance for SGD. This potentially indicates that a generic regularization strategy may not suffice for overparameterized models relatively compared to SAM.

## G.3 RESULTS ON SAM UNDER LINEARIZED REGIME

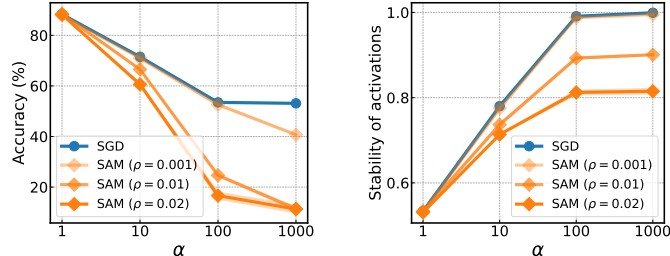

Figure 21: Effect of linearization on SAM. Here, $\alpha$ controls the degree of linearization. High linearization does not yield improvement, and in fact, SAM ($\rho = 0.001$) underperforms SGD in the linearized regime (left), although both achieve effective linearization at $\alpha = 1000$ with stability close to 1 (right).

Recent studies suggest that highly overparameterized models can behave like linearized networks (Jacot et al., 2018), while such implicit linearization phenomenon can coincide independently of overparameterization (Chizat et al., 2019). One might wonder if the increased effectiveness of SAM directly comes from the overparameterization itself or is rather due to linearization. To verify, we reproduce experiments in Chizat et al. (2019) and see how SAM performs in the linearized regimes

while fixing the number of parameters. Specifically, we train VGG-11 (Simonyan & Zisserman, 2015) on the Cifar-10 with the $\alpha$-scaled squared loss $L(x, y) = \|f(x) - y/\alpha\|^2$ and use the centered model whose initial output is set to 0. Here, a large value of $\alpha$ leads to a higher degree of linearization of the models.

The results are reported in Figure 21. We observe that SAM underperforms SGD in the linearized regimes; while SAM ($\rho = 0.001$) and SGD both achieve effective linearization at $\alpha = 1000$, SAM underperforms SGD by more than 10%. This indicates that linearization is not the main factor, and overparameterization itself is what leads to the improvement of SAM in previous experiments.

## H    EMPIRICAL MEASUREMENT OF LIPSCHITZ SMOOTHNESS AND PL CONSTANTS

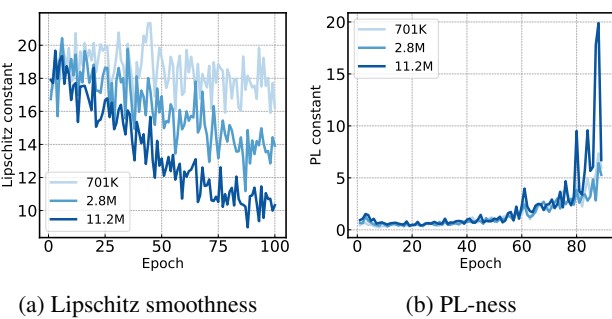

(a) Lipschitz smoothness                 (b) PL-ness

Figure 22: The empirical measurement of Lipschitz smoothness (a) and PL-ness (b) for CIFAR-10 and ResNet-18. The Lipschitz smoothness becomes smaller and PL constant becomes larger as the model size increaess.

We measure the empirical Lipschitz smoothness constant and PL constant based on Zhang et al. (2019). Specifically, the empirical smoothness $\hat{\beta}(x_k)$ and empirical PL constant $\hat{\alpha}(x_k)$ at iteration $k$ is computed as follows:

$$\hat{\beta}(x_k) = \max_{\gamma \in \{\delta, 2\delta, ..., 1\}} \frac{\|\nabla f(x_k + \gamma \mathbf{d}) - \nabla f(x_k)\|_2}{\|\gamma \mathbf{d}\|_2}, \tag{8}$$

$$\hat{\alpha}(x_k) = \min_{\gamma \in \{\delta, 2\delta, ..., 1\}} \frac{\|\nabla f(x_k + \gamma \mathbf{d})\|_2^2}{f(x_k + \gamma \mathbf{d}) - f(x^\star)}. \tag{9}$$

where $\mathbf{d} = x_{k+1} - x_k$ and $\delta \in (0, 1)$ where we choose $\delta = 0.1$. We measure these quantities at the end of every epoch throughout training. The results are shown in Figure 22.

## I    PROOF OF THEOREM 6.3

Here, we provide the detailed proof of Theorem 6.3.

We first define a linearized stochastic SAM, which is derived from applying first-order Taylor approximation on a stochastic SAM update given as follows:

**Definition I.1.** (Linearized stochastic SAM) We define a linearized stochastic SAM as

$$x_{t+1} = x_t - \eta H_{\xi_t}(x_{t+1/2} - x^\star), \tag{10}$$

where $x_{t+1/2} = x_t + \rho H_{\xi_t}(x_t - x^\star)$ is the linearized ascent step and $H_{\xi_t}$ is the Hessian estimation at step $t$.

This actually corresponds to using SAM for the quadratic approximation of $f$ near $x^\star$, and we use this fact in the experiment setup. We assume without loss of generality that the fixed point $x^\star$ satisfies $x^\star = 0$.

Then, we are ready to present the proof of Theorem 6.3. Our goal is to derive a bound of the form $\mathbb{E}\|x_t\|^2 \leq C\|x_0\|^2$. We first apply (10) to $\mathbb{E}\left[\|x_{t+1}\|^2 \mid x_t\right]$ and continue expanding the terms as follows:

$$
\begin{aligned}
\mathbb{E}\left[\|x_{t+1}^2\| \mid x_t\right] &= \mathbb{E}\|x_t - \eta H_{\xi_t}(x_t + \rho H_{\xi_t} x_t)\|^2 \\
&= x_t^\top \mathbb{E}\left[\left(I - \eta H_{\xi_t} - \eta\rho H_{\xi_t}^2\right)^2 \,\Big|\, x_t\right] x_t \\
&= x_t^\top \mathbb{E}\left[I - 2\eta(H_{\xi_t} + \rho H_{\xi_t}^2) + \eta^2\left(H_{\xi_t} + \rho H_{\xi_t}^2\right)^2 \,\Big|\, x_t\right] x_t \\
&= x_t^\top \mathbb{E}\left[I - 2\eta(H_{\xi_t} + \rho H_{\xi_t}^2) + \eta^2\left(H_{\xi_t}^2 + 2\rho H_{\xi_t}^3 + \rho^2 H_{\xi_t}^4\right) \,\Big|\, x_t\right] x_t \\
&= x_t^\top \mathbb{E}\left[I - 2\eta H_{\xi_t} + \eta(\eta - 2\rho)H_{\xi_t}^2 + 2\eta^2\rho H_{\xi_t}^3 + \eta^2\rho^2 H_{\xi_t}^4 \,\Big|\, x_t\right] x_t \\
&= x_t^\top \left(I - 2\eta H + \eta(\eta - 2\rho)\mathbb{E}H_{\xi_t}^2 + 2\eta^2\rho\mathbb{E}H_{\xi_t}^3 + \eta^2\rho^2\mathbb{E}H_{\xi_t}^4\right) x_t \\
&= x_t^\top \left(I - 2\eta H + \eta(\eta - 2\rho)H^2 + 2\eta^2\rho H^3 + \eta^2\rho^2 H^4 \right. \\
&\qquad \left. + \eta(\eta - 2\rho)(\mathbb{E}H_{\xi_t}^2 - H^2) + 2\eta^2\rho(\mathbb{E}H_{\xi_t}^3 - H^3) + \eta^2\rho^2(\mathbb{E}H_{\xi_t}^4 - H^4)\right) x_t \\
&= x_t^\top \left(\left(I - \eta H - \eta\rho H^2\right)^2 \right. \\
&\qquad \left. + \eta(\eta - 2\rho)(\mathbb{E}H_{\xi_t}^2 - H^2) + 2\eta^2\rho(\mathbb{E}H_{\xi_t}^3 - H^3) + \eta^2\rho^2(\mathbb{E}H_{\xi_t}^4 - H^4)\right) x_t
\end{aligned}
$$

Since $x^\top A x \leq \lambda_{\max}(A)\|x\|^2$ always holds for any $x$ and any matrix $A$ with the maximum eigenvalue $\lambda_{\max}(A)$, applying this inequality and taking the total expectation gives the following;

$$
\begin{aligned}
\mathbb{E}\left[\|x_{t+1}\|^2\right] \leq \lambda_{\max}\bigg( &\left(I - \eta H - \eta\rho H^2\right)^2 + \eta(\eta - 2\rho)(\mathbb{E}H_\xi^2 - H^2) \\
&+ 2\eta^2\rho(\mathbb{E}H_\xi^3 - H^3) + \eta^2\rho^2(\mathbb{E}H_\xi^4 - H^4)\bigg)\mathbb{E}\left[\|x_t\|^2\right].
\end{aligned}
$$

Recursively applying this bound gives

$$
\begin{aligned}
\mathbb{E}\|x_t\|^2 \leq \lambda_{\max}\bigg( &\left(I - \eta H - \eta\rho H^2\right)^2 + \eta(\eta - 2\rho)(\mathbb{E}H_\xi^2 - H^2) \\
&+ 2\eta^2\rho(\mathbb{E}H_\xi^3 - H^3) + \eta^2\rho^2(\mathbb{E}H_\xi^4 - H^4)\bigg)^t \|x_0\|^2.
\end{aligned}
$$

Here, we can see that $x^\star$ is linearly stable if

$$
\begin{aligned}
\lambda_{\max}\bigg( &(I - \eta H - \eta\rho H^2)^2 \\
&+ \eta(\eta - 2\rho)(\mathbb{E}H_\xi^2 - H^2) + 2\eta^2\rho(\mathbb{E}H_\xi^3 - H^3) + \eta^2\rho^2(\mathbb{E}H_\xi^4 - H^4)\bigg) \leq 1.
\end{aligned}
$$

$\square$

## J  PROOF OF THEOREM 6.6

In this section, we show that a stochastic SAM converges linearly under an overparameterized regime. To put into perspective, this is the rate of convergence of gradient descent for a family of functions satisfying the PL-condition and smoothness assumptions (Karimi et al., 2016). We first make several remarks on this result below.

- Crucially, this result shows that with overparameterization, a stochastic SAM can converge as fast as the deterministic gradient method at a linear convergence rate. It is much faster than the well-known sublinear rate of $\mathcal{O}(1/t)$ for SAM (Andriushchenko & Flammarion, 2022).

- When $\rho = 0$, we recover the well-known convergence rate for SGD in interpolated regime (Bassily et al., 2018).

- This result does not require the bounded variance assumption (Andriushchenko & Flammarion, 2022) since the interpolation provides necessary guarantees. This suggests that overparameterization can ease the convergence of SAM.

We prove the convergence for an unnormalized mini-batch SAM given as

$$x_{t+1} = x_t - \eta g_t^B(x_t + \rho g_t^B(x_t)),$$

where $g_t^B(x) = \frac{1}{B} \sum_{i \in I_t^B} \nabla f_i(x)$ and $I_t^B \subseteq \{1, ..., n\}$ is a set of indices for data points in the mini-batch of size $B$ sampled at step $t$. This is a more general stochastic variant of SAM where a stochastic SAM in Section 6.2 is a particular case of a mini-batch SAM with mini-batch size $B = 1$.

We first make the following assumptions:

**(A1)** ($\beta$-smoothness of $f_i$). *There exists $\beta > 0$ s.t. $\|\nabla f_i(x) - \nabla f_i(y)\| \leq \beta \|x - y\|$ for all $x, y \in \mathbb{R}^d$,*

**(A2)** ($\lambda$-smoothness of $f$). *There exists $\lambda > 0$ such that $\|\nabla f(x) - \nabla f(y)\| \leq \lambda \|x - y\|$ for all $x, y \in \mathbb{R}^d$,*

**(A3)** ($\alpha$-PLness of $f$). *There exists $\alpha > 0$ s.t. $\|\nabla f(x)\|^2 \geq \alpha(f(x) - f(x^\star))$ for all $w, v \in \mathbb{R}^d$,*

**(A4)** (Interpolation). *If $f(x^\star) = 0$ and $\nabla f(x^\star) = 0$, then $f_i(x^\star) = 0$ and $\nabla f_i(x^\star) = 0$ for $i = 1, \ldots, n$, where $n$ is the number of training data points.*

Before we prove the main theorem, we first introduce two lemma important to the proof.

**Lemma J.1.** *Suppose that Assumption (A1) holds. Then*

$$\langle \nabla f_i(x_{t+1/2}), \nabla f(x_t) \rangle \geq \langle \nabla f_i(x_t), \nabla f(x_t) \rangle - \frac{\beta \rho}{2} \|\nabla f_i(x_t)\|^2 - \frac{\beta \rho}{2} \|\nabla f(x_t)\|^2, \quad (11)$$

*where $x_{t+1/2} = x_t + \rho \nabla f_i(x_t)$.*

This lemma shows how well a stochastic SAM gradient $\nabla f_i(x_{t+1/2})$ aligns with the true gradient $\nabla f(x_t)$. The two gradients become less aligned as $\beta$ and $\rho$ grow bigger, *i.e.* for sharper landscape and larger perturbation size.

*Proof.* We first add and subtract $\nabla f_i(x_t)$ on the left side of the inner product

$$\langle \nabla f_i(x_{t+1/2}), \nabla f(x_t) \rangle = \underbrace{\langle \nabla f_i(x_{t+1/2}) - \nabla f_i(x_t), \nabla f(x_t) \rangle}_{\tau_1} + \langle \nabla f_i(x_t), \nabla f(x_t) \rangle. \quad (12)$$

We here bound the term $\tau_1$ so that it becomes an equality when $\rho = 0$. To achieve this, we start with the following binomial square, which is trivially lower bounded by 0.

$$0 \leq \frac{1}{2} \|\nabla f_i(x_{t+1/2}) - \nabla f_i(x_t) + \beta \rho \nabla f(x_t)\|^2$$

We then expand the above binomial square so that the term containing $\tau_1$ appears.

$$0 \leq \frac{1}{2} \|\nabla f_i(x_{t+1/2}) - \nabla f_i(x_t)\|^2 + \underbrace{\langle \nabla f_i(x_{t+1/2}) - \nabla f_i(x_t), \beta \rho \nabla f(x_t) \rangle}_{\beta \rho \tau_1} + \frac{1}{2} \|\beta \rho \nabla f(x_t)\|^2$$

We subtract the term $\beta \rho \tau_1$ on both sides of the inequality which gives

$$-\langle \nabla f_i(x_{t+1/2}) - \nabla f_i(x_t), \beta \rho \nabla f(x_t) \rangle \leq \frac{1}{2} \|\nabla f_i(x_{t+1/2}) - \nabla f_i(x_t)\|^2 + \frac{\beta^2 \rho^2}{2} \|\nabla f(x_t)\|^2.$$

Then we upper bound the right-hand side using the Assumption (A1):

$$-\langle \nabla f_i(x_{t+1/2}) - \nabla f_i(x_t) \,,\, \beta\rho\nabla f(x_t)\rangle \leq \frac{\beta^2}{2}\|x_{t+1/2} - x\|^2 + \frac{\beta^2\rho^2}{2}\|\nabla f(x_t)\|^2$$
$$= \frac{\beta^2\rho^2}{2}\|\nabla f_i(x_t)\|^2 + \frac{\beta^2\rho^2}{2}\|\nabla f(x_t)\|^2.$$

We divide both sides with $\beta\rho$, obtaining:

$$-\langle \nabla f_i(x_{t+1/2}) - \nabla f_i(x_t), \nabla f(x_t)\rangle \leq \frac{\beta\rho}{2}\|\nabla f_i(x_t)\|^2 + \frac{\beta\rho}{2}\|\nabla f(x_t)\|^2.$$

Applying this to (12) gives the bound in the lemma statement. $\qquad\square$

**Lemma J.2.** *Suppose that Assumption (A1) holds. Then*

$$\left\|\nabla f_i(x_{t+1/2})\right\|^2 \leq (\beta\rho + 1)^2\|\nabla f_i(x_t)\|^2, \tag{13}$$

*where $x_{t+1/2} = x_t + \rho\nabla f_i(x_t)$.*

This second lemma shows that the norm of a stochastic SAM gradient is bounded by the norm of the stochastic gradient. Similar to the Lemma J.1, as $\beta$ and $\rho$ grow bigger the norm for a stochastic SAM gradient can become larger than the norm of the true gradient.

*Proof.* We use the following binomial squares:

$$\|\nabla f_i(x_{t+1/2})\|^2$$
$$= \|\nabla f_i(x_{t+1/2}) - \nabla f_i(x_t)\|^2 + 2\langle \nabla f_i(x_{t+1/2}) - \nabla f_i(x_t), \nabla f_i(x_t)\rangle + \|\nabla f_i(x_t)\|^2.$$

We bound the right-hand side using Cauchy-Schwarz inequality and Assumption (A1), which gives

$$\left\|\nabla f_i(x_{t+1/2})\right\|^2$$
$$= \|\nabla f_i(x_{t+1/2}) - \nabla f_i(x_t)\|^2 + 2\langle \nabla f_i(x_{t+1/2}) - \nabla f_i(x_t), \nabla f_i(x_t)\rangle + \|\nabla f_i(x_t)\|^2$$
$$\underset{\text{C.S.}}{\leq} \|\nabla f_i(x_{t+1/2}) - \nabla f_i(x_t)\|^2 + 2\|\nabla f_i(x_{t+1/2}) - \nabla f_i(x_t)\|\|\nabla f_i(x_t)\| + \|\nabla f_i(x_t)\|^2$$
$$\underset{\text{(A1)}}{\leq} \beta^2\|x_{t+1/2} - x_t\|^2 + 2\beta\|x_{t+1/2} - x_t\|\|\nabla f_i(x_t)\| + \|\nabla f_i(x_t)\|^2$$
$$= \beta^2\rho^2\|\nabla f_i(x_t)\|^2 + 2\beta\rho\|\nabla f_i(x_t)\|^2 + \|\nabla f_i(x_t)\|^2$$
$$= (\beta\rho + 1)^2\|\nabla f_i(x_t)\|^2$$

$\qquad\square$

These two lemmas essentially show how similar a stochastic SAM gradient is to the stochastic gradient, where the two become more similar as $\beta$ and $\rho$ decrease, which aligns well with our intuition. Using Lemma J.1 and J.2, we provide the convergence result in the following theorem.

**Theorem J.3.** *Suppose that Assumptions (A1-4) holds. For any mini-batch size $B \in \mathbb{N}$ and $\rho \leq \frac{1}{(\beta/\alpha+1/2)\beta}$, unnormalized mini-batch SAM with constant step size $\eta_B^\star \overset{def}{=} \frac{1-(\kappa_B+1/2)\beta\rho}{2\lambda\kappa_B(\beta\rho+1)^2}$ gives the following guarantee at step $t$:*

$$\mathbb{E}_{x_t}[f(x_t)] \leq \left(1 - \frac{\alpha\,\eta_B^\star}{2}\left(1 - \left(\kappa_B + \frac{1}{2}\right)\beta\rho\right)\right)^t f(x_0),$$

*where $\kappa_B = \frac{1}{B}\left(\frac{B-1}{2} + \frac{\beta}{\alpha}\right)$.*

This theorem states that mini-batch SAM converges at a linear rate under overparameterization.

*Proof.* Proof can be outlined in 3 steps.

---

**step 1.** Handle terms containing mini-batch SAM gradient $g_t^B(x_t + \rho g_t^B(x_t))$ using bounds from **(A1)**.

**step 2.** Take conditional expectation $\mathbb{E}[\,\cdot\,|x_t]$ and substitute expectation of function of mini-batch gradient $g_t^B$ with terms containing $\|\nabla f(x_t)\|$ and $\mathbb{E}\left[\|\nabla f_i(x_t)\|^2 \mid x_t\right]$.

**step 3.** Bound the two terms from **step 2**, one using Assumptions **(A1)** and **(A4)** and the other using Assumption **(A3)** and **(A4)** which results in all the terms to contain $f(x_t)$. Then finally we take total expectations to derive the final runtime bound.

---

We start from the quadratic upper bound derived from Assumption **(A2)**;

$$f(x_{t+1}) \leq f(x_t) + \langle \nabla f(x_t),\ x_{t+1} - x_t \rangle + \frac{\lambda}{2}\|x_{t+1} - x_t\|^2.$$

Applying mini-batch SAM update, we then have

$$f(x_t) - f(x_{t+1}) \geq \eta \left\langle \nabla f(x_t),\ g_t^B(x_{t+1/2}) \right\rangle - \frac{\eta^2 \lambda}{2} \left\| g_t^B(x_{t+1/2}) \right\|^2,$$

where $x_{t+1/2} = x_t + \rho g_t^B(x_t)$.

$\boxed{\textbf{step 1.}}$  We can see that there are two terms that contain a mini-batch SAM gradient $g_t^B(x_{t+1/2})$. We see that each can be bounded directly using Lemma J.1 and J.2, which gives

$$
\begin{aligned}
&f(x_t) - f(x_{t+1}) \\
&\geq \eta \left( \langle g_t^B(x_t), \nabla f(x_t)\rangle - \frac{\beta\rho}{2}\|g_t^B(x_t)\|^2 - \frac{\beta\rho}{2}\|\nabla f(x_t)\|^2 \right) \\
&\quad - \frac{\eta^2\lambda}{2}(\beta\rho+1)^2\,\|g_t^B(x_t)\|^2 \\
&= \eta\langle g_t^B(x_t), \nabla f(x_t)\rangle - \frac{\eta\beta\rho}{2}\|\nabla f(x_t)\|^2 - \frac{\eta}{2}\left(\eta\lambda(\beta\rho+1)^2 + \beta\rho\right)\,\|g_t^B(x_t)\|^2.
\end{aligned}
$$

$\boxed{\textbf{step 2.}}$  Now we apply $\mathbb{E}[\,\cdot\,|x_t]$ to all the terms.

$$
\begin{aligned}
&\mathbb{E}\big[f(x_t) - f(x_{t+1}) \mid x_t\big] \\
&= f(x_t) - \mathbb{E}\big[f(x_{t+1}) \mid x_t\big] \\
&\geq \eta\mathbb{E}\left[\langle g_t^B(x_t), \nabla f(x_t)\rangle \mid x_t\right] - \frac{\eta\beta\rho}{2}\mathbb{E}\big[\|\nabla f(x_t)\|^2 \mid x_t\big] \\
&\quad - \frac{\eta}{2}\left(\eta\lambda(\beta\rho+1)^2 + \beta\rho\right)\mathbb{E}\left[\|g_t^B(x_t)\|^2 \mid x_t\right] \\
&= \eta\left(1 - \frac{\beta\rho}{2}\right)\|\nabla f(x_t)\|^2 - \frac{\eta}{2}\left(\eta\lambda(\beta\rho+1)^2 + \beta\rho\right)\mathbb{E}\left[\|g_t^B(x_t)\|^2 \mid x_t\right].
\end{aligned}
$$

Here we expand the term $\mathbb{E}\left[\|g_t^B(x_t)\|^2 \mid x_t\right]$ by expanding the mini-batched function into individual function estimators as follows.

$$\mathbb{E}_{g_t^B}\left[\|g_t^B(x_t)\|^2 \mid x_t\right]$$

$$= \mathbb{E}_{I_t^B}\left[\left\langle \frac{1}{B}\sum_{i\in I_t^B}\nabla f_i(x_t) , \frac{1}{B}\sum_{j\in I_t^B}\nabla f_j(x_t)\right\rangle \mid x_t\right]$$

$$= \frac{1}{B^2}\left\{\sum_{i\in I_t^B}\mathbb{E}_{f_i}\left[\|\nabla f_i(x_t)\|^2 \mid x_t\right] + \sum_{i\in I_t^B}\sum_{\substack{j\in I_t^B\\(j\neq i)}}\mathbb{E}_{f_i,f_j}\left[\langle\nabla f_i(x_t),\nabla f_j(x_t)\rangle \mid x_t\right]\right\} \tag{14}$$

$$= \frac{1}{B}\mathbb{E}\left[\|\nabla f_i(x_t)\|^2 \mid x_t\right] + \frac{B-1}{B}\|\nabla f(x_t)\|^2 .$$

Using (14), we get

$$f(x_t) - \mathbb{E}\left[f(x_{t+1}) \mid x_t\right] \tag{15}$$

$$\geq \eta\left(1 - \frac{\beta\rho}{2}\right)\|\nabla f(x_t)\|^2$$

$$- \frac{\eta}{2}\left(\eta\lambda(\beta\rho+1)^2 + \beta\rho\right)\left(\frac{1}{B}\mathbb{E}\left[\|\nabla f_i(x_t)\|^2 \mid x_t\right] + \frac{B-1}{B}\|\nabla f(x_t)\|^2\right)$$

$$= \eta\left(\left(1 - \frac{\beta\rho}{2}\right) - \frac{B-1}{2B}\left(\eta\lambda(\beta\rho+1)^2 + \beta\rho\right)\right)\|\nabla f(x_t)\|^2$$

$$- \frac{\eta}{2B}\left(\eta\lambda(\beta\rho+1)^2 + \beta\rho\right)\mathbb{E}\left[\|\nabla f_i(x_t)\|^2 \mid x_t\right]. \tag{16}$$

**step 3.** In this step, we bound the two terms and take the total expectation to derive the final runtime bound.

We first derive a bound for $\mathbb{E}\left[\|\nabla f_i(x_t)\|^2 \mid x_t\right]$. We start from the following bound derived from Assumption **(A1)**:

$$\|\nabla f_i(x_t) - \nabla f_i(x^\star)\|^2 \leq 2\beta(f_i(x_t) - f_i(x^\star)).$$

By Assumption **(A4)**, this reduces to

$$\|\nabla f_i(x_t)\|^2 \leq 2\beta f_i(x_t).$$

Applying this to (16) gives

$$f(x_t) - \mathbb{E}\left[f(x_{t+1}) \mid x_t\right] \geq \eta\left(\left(1 - \frac{\beta\rho}{2}\right) - \frac{B-1}{2B}\left(\eta\lambda(\beta\rho+1)^2 + \beta\rho\right)\right)\|\nabla f(x_t)\|^2$$

$$- \frac{\eta\beta}{B}\left(\eta\lambda(\beta\rho+1)^2 + \beta\rho\right)\mathbb{E}[f_i(x_t)|x_t]$$

$$= \eta\underbrace{\left(\left(1 - \frac{\beta\rho}{2}\right) - \frac{B-1}{2B}\left(\eta\lambda(\beta\rho+1)^2 + \beta\rho\right)\right)}_{\tau_2}\|\nabla f(x_t)\|^2$$

$$- \frac{\eta\beta}{B}\left(\eta\lambda(\beta\rho+1)^2 + \beta\rho\right)f(x_t). \tag{17}$$

Next, to bound $\|\nabla f(x_t)\|^2$, we use the following bound derived from applying $f(x^*) = 0$ from **(A4)** to **(A3)**:

$$\|\nabla f(x)\|^2 \geq \alpha f(x). \tag{18}$$

Assuming $\tau_2 \geq 0$ which we provide a sufficient condition at the end of the proof, we apply (18) to (17) which gives

$$f(x_t) - \mathbb{E}\big[f(x_{t+1}) \mid x_t\big]$$

$$\geq \eta\alpha\left(\left(1 - \frac{\beta\rho}{2}\right) - \frac{B-1}{2B}\left(\eta\lambda(\beta\rho+1)^2 + \beta\rho\right)\right)f(x_t) - \frac{\eta\beta}{B}\left(\eta\lambda(\beta\rho+1)^2 + \beta\rho\right)f(x_t)$$

$$= \eta\left(\alpha - \alpha\left(\underbrace{\frac{1}{B}\left(\frac{B-1}{2} + \frac{\beta}{\alpha}\right)}_{\kappa_B} + \frac{1}{2}\right)\beta\rho - \eta(\beta\rho+1)^2\underbrace{\frac{\lambda}{B}\left(\alpha\frac{B-1}{2} + \beta\right)}_{\lambda\alpha\kappa_B}\right)f(x_t)$$

$$= \eta\alpha\left(1 - \left(\kappa_B + \frac{1}{2}\right)\beta\rho - \eta\lambda(\beta\rho+1)^2\kappa_B\right)f(x_t).$$

Hence, we get

$$\mathbb{E}\big[f(x_{t+1}) \mid x_t\big] \leq \left(1 - \eta\alpha\left(1 - \left(\kappa_B + \frac{1}{2}\right)\beta\rho\right) + \eta^2\alpha\lambda(\beta\rho+1)^2\kappa_B\right)f(x_t).$$

Applying total expectation on both sides gives

$$\mathbb{E}[f(x_{t+1})] \leq \left(1 - \eta\alpha\left(1 - \left(\kappa_B + \frac{1}{2}\right)\beta\rho\right) + \eta^2\alpha\lambda(\beta\rho+1)^2\kappa_B\right)\mathbb{E}[f(x_t)]. \tag{19}$$

Optimizing the multiplicative term in (19) with respect to $\eta$ gives $\eta = \frac{1-(\kappa_B+1/2)\beta\rho}{2\lambda\kappa_B(\beta\rho+1)^2}$, which is $\eta_B^\star$ in the theorem statement. With assumption of $\rho \leq \frac{1}{(\beta/\alpha+1/2)\beta}$ so that we have $\eta_B^\star \geq 0$, applying this to (19) gives

$$\mathbb{E}\left[f(x_{t+1})\right] \leq \left(1 - \frac{\alpha\,\eta_B^\star}{2}\left(1 - \left(\kappa_B + \frac{1}{2}\right)\beta\rho\right)\right)\mathbb{E}\left[f(x_t)\right],$$

which provides the desired convergence rate.

Last but not least, we calculate the upper bound for $\rho$ to satisfy the assumption $\tau_2 \geq 0$ by substituting $\eta$ for $\eta_B^\star$ in $\tau_2$, yielding $\rho \leq \frac{2B\kappa_B+2\beta/\alpha}{(2B-1)\kappa_B+\beta/\alpha}\frac{1}{\beta}$. Minimizing this upper bound with respect to $B$ gives $\rho \leq \frac{1}{\beta}$, which is a looser bound than $\rho \leq \frac{1}{(\beta/\alpha+1/2)\beta}$. $\qquad\square$

## K    TEST ERROR OF SAM CAN DECREASE WITH OVERPARAMETERIZATION

Recent works have shown that overparameterization can even improve generalization both empirically and theoretically (Neyshabur et al., 2017; Brutzkus & Globerson, 2019). Here, we present that overparameterization also improves generalization for SAM in the sense that test error can decrease with larger network widths (and thus more parameters).

We follow the same setting of Allen-Zhu et al. (2019). Specifically, we consider a risk minimization over some unknown data distribution $\mathcal{D}$ using a one-hidden-layer ReLU network with a smooth convex loss function (*e.g.*, cross entropy). The network is assumed to be initialized with Gaussian and take bounded inputs. Then, we characterize a generalization property of a stochastic SAM as below.

**Theorem K.1.** *(Informal) Suppose we train a network having $m$ hidden neurons with training data sampled from $\mathcal{D}$. Then, for every $\varepsilon$ in some open interval, there exists $M \propto 1/\varepsilon$ such that for every $m \geq M$, with appropriate values of $\eta, \rho, T$, a stochastic SAM gives the following guarantee on the test loss with high probability:*

$$\mathbb{E}_{x_0,\cdots,x_{T-1}}\left[\frac{1}{T}\sum_{t=0}^{T-1}\mathbb{E}_{\mathcal{D}}[f(x_t)]\right] \leq \varepsilon.$$

We present a formal version of the theorem and its proof in Appendix L.

This result suggests that to achieve $\varepsilon$-test accuracy from running $T$ iterations of SAM requires a minimum width $M$ proportional to $1/\varepsilon$. This indicates that a network with a larger width can achieve a lower test error, and hence, overparameterization can improve generalization for SAM.

**Experiment** We support this result empirically on synthetic data for a simple regression task. Specifically, following the setup of Allen-Zhu et al. (2019), we train 2-layer ReLU networks with synthetic data. Here, each element of the input $x = (x_1, x_2, x_3, x_4) \in \mathbb{R}^4$ for synthetic data is sampled from random Gaussian distribution and then normalized to satisfy $\|x\|_2 = 1$, and target $y$ is calculated as $y = (\sin(3x_1) + \sin(3x_2) + \sin(3x_3) - 2)^2 \cdot \cos(7x_4)$. The weights and biases of the first layer are initialized from $\mathcal{N}(0, 1/m)$ where $m$ is the number of hidden neurons, and the weights of the second layer are initialized from $\mathcal{N}(0, 1)$. We only train the weights of the first layer for 800 epochs, while the biases of the first layer and the weights of the second layer are frozen to initialized values. We use 1000 and 5000 data points for training and testing respectively. We use a batch size of 50 without weight decay and decay learning rate by 0.1 after 50% of the total epochs. We perform the grid search over learning rate and $\rho$ from $\{10^{-k} | 2 \leq k \leq 7\}$ and $\{10^{-k} | 1 \leq k \leq 5\}$ respectively.

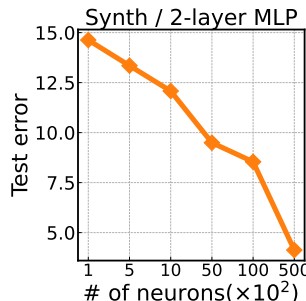

Figure 23: Generalization of SAM. Test error keeps on decreasing with a larger number of neurons.

## L    PROOF OF THEOREM K.1

In this section, we provide the formal version of Theorem K.1 and its proof.

### L.1    NOTATION AND SETUP

Throughout this section, we use the same notations and setups as Allen-Zhu et al. (2019). We remark that the notations are different from those used in Appendices I to K.

First, let us assume the unknown data distribution $\mathcal{D}$ where each data $z = (x, y)$ consists of the input $x \in \mathbb{R}^d$ and the corresponding label $y \in \mathcal{Y}$. We also assume, without loss of generality, that $\|x\|_2 = 1$ and $x_d = 1/2$. The loss function $L : \mathbb{R}^k \times \mathcal{Y} \to \mathbb{R}$ is assumed to be non-negative, convex, 1-Lipschitz continuous, and 1-smooth with respect to its first argument.

Next, we define the target network $F^* = (f_1^*, \cdots, f_k^*) : \mathbb{R}^d \to \mathbb{R}^k$ as

$$f_r^*(x) \stackrel{\text{def}}{=} \sum_{i=1}^{p} a_{r,i}^* \phi_i(\langle w_{1,i}^*, x \rangle) \langle w_{2,i}^*, x \rangle \tag{20}$$

where each $\phi_i : \mathbb{R} \to \mathbb{R}$ is an infinite-order smooth function. Here, we assume that $\|w_{1,i}^*\|_2 = \|w_{2,i}^*\|_2 = 1, |a_{r,i}^*| \leq 1$ hold for all $i \in \{1, \cdots, p\}$. We denote the sample and network complexity of $\phi$ as $\mathfrak{C}_{\mathfrak{s}}$ and $\mathfrak{C}_{\epsilon}$ respectively (see Section 2 of Allen-Zhu et al. (2019) for the formal definitions). Suppose we have a concept class $\mathcal{C}$ that consists of all functions $F^*$ with bounded number of parameters $p$ and complexity $\mathfrak{C}$. We also denote the population risk achieved by the best target function $F^*$ in this concept class as $\mathsf{OPT}$, *i.e.*, $\mathsf{OPT} = \min_{F^* \in \mathcal{C}} \mathbb{E}_{(x,y) \sim \mathcal{D}}[L(F^*(x), y]$

Then, we define the learner network $F = (f_1, \cdots, f_k) : \mathbb{R}^d \to \mathbb{R}^k$ as below.

$$f_r(x) \stackrel{\text{def}}{=} \sum_{i=1}^{m} a_{r,i}^{(0)} \text{ReLU}(\langle w_i, x \rangle + b_i^{(0)}). \tag{21}$$

Note that the learner network is a 2-layer ReLU network with $m$ neurons. We train the network with $n$ sampled data sampled from $\mathcal{D}$ and denote it as $\mathcal{Z} = \{z_1, \cdots, z_N\}$. We only train the weights $W = (w_1, \cdots, w_m) \in \mathbb{R}^{m \times d}$ and freeze the values of $a, b$ during the training. We denote the initial value of the weight and its value at time $t$ as $W^{(0)}$ and $W^{(0)} + W_t$ respectively. Each element of $W^{(0)}$ and $b^{(0)}$ are initialized from $\mathcal{N}(0, 1/m)$ while each element of $a_r^{(0)}$ are initialized from $\mathcal{N}(0, \varepsilon_a^2)$ for some fixed $\varepsilon_a \in (0, 1]$. At each step $t$, we sample a single data point $z = (x, y)$ from $\mathcal{Z}$ and update $W$ using un-normalized version of SAM:

$$W_{t+1} = W_t - \eta \nabla L(F(x; W^{(0)} + W_{t+1/2}), y)$$
$$= W_t - \eta \nabla L(F(x; W^{(0)} + \rho \nabla L(F(x; W^{(0)} + W_t), y)), y). \tag{22}$$

## L.2 FORMAL THEOREM

Now, we are ready to present the formal version of Theorem K.1 below.

**Theorem L.1.** *(SAM version of Theorem 1 in Allen-Zhu et al. (2019)) For every $\varepsilon \in \left(0, \frac{1}{pk\mathfrak{C}_s(\phi,1)}\right)$, there exists $M_0 = \mathsf{poly}(\mathfrak{C}_\epsilon(\phi,1), 1/\varepsilon)$ and $N_0 = \mathsf{poly}(\mathfrak{C}_s(\phi,1), 1/\varepsilon)$ such that for every $m \geq M_0$ and every $N \geq \widetilde{\Omega}(N_0)$, by choosing $\varepsilon_a = \varepsilon/\widetilde{\Theta}(1)$ for the initialization and $\eta = \widetilde{\Theta}(\frac{1}{\varepsilon km})$, $\rho = \widetilde{\Theta}(\frac{1}{\varepsilon^3 km^3})$, $T = \widetilde{\Theta}\left(\frac{(\mathfrak{C}_s(\phi,1))^2 \cdot k^3 p^2}{\varepsilon^2}\right)$, running $T$ iterations of stochastic SAM defined in Equation (22) gives the following generalization bound with high probability over the random initialization.*

$$\mathbb{E}_{SAM}\left[\frac{1}{T}\sum_{t=0}^{T-1}\mathbb{E}_{(x,y)\sim\mathcal{D}}L(F(x;W^{(0)}+W_t),y)\right] \leq \mathsf{OPT} + \varepsilon. \tag{23}$$

Here, the notation of $\widetilde{O}(\cdot)$ ignores the factor of $\mathsf{polylog}(m)$.

## L.3 PROOF OF THEOREM L.1

We here present the proof of Theorem L.1.

First, note that we can directly use the algorithm-independent part from Allen-Zhu et al. (2019). Thus, it is sufficient to show that the similar version of Lemma B.4 in Allen-Zhu et al. (2019) also holds for SAM.

We first define the function $G = (g_1, \cdots, g_k) : \mathbb{R}^d \to \mathbb{R}^k$ as similar to Allen-Zhu et al. (2019).

$$g_r(x;W_t) \overset{\text{def}}{=} \sum_{i=1}^{m} a_{r,i}^{(0)}(\langle w_i^{(t)}, x\rangle + b_i^{(0)})\mathbb{1}[\langle w_i^{(0)}, x\rangle + b_i^{(0)} \geq 0]. \tag{24}$$

Then, the following corollary holds for a stochastic SAM from Lemma B.3 of Allen-Zhu et al. (2019). The corollary presents an upper bound on the norm of differences between $\frac{\partial}{\partial W}L(F(\cdot),y)$ and $\frac{\partial}{\partial W}L(G(\cdot),y)$.

**Corollary L.2.** *(SAM version of Lemma B.3 in Allen-Zhu et al. (2019)) Let $\tau = \varepsilon_a(\eta + \rho)t$. Then, for every $x$ satisfying $\|x\|_2 = 1$, and for every time step $t \geq 1$, the following are satisfied with high probability over the random initialization.*
*(a) For every $r \in [k]$,*

$$\left|f_r(x;W^{(0)}+W_t) - g_r(x;W^{(0)}+W_t)\right| = \widetilde{O}(\varepsilon_a k\tau^2 m^{3/2})$$

*(b) For every $y \in \mathcal{Y}$,*

$$\left\|\frac{\partial}{\partial W}L(F(x;W^{(0)}+W_t),y) - \frac{\partial}{\partial W}L(G(x;W^{(0)}+W_t),y)\right\|_{2,1} \leq \widetilde{O}(\varepsilon_a k\tau m^{3/2} + \varepsilon_a^2 k^2 \tau^2 m^{5/2})$$
$$\tag{25}$$

Next, we present the key lemma integral to our proof. The part $(c)$ will be directly used in the proof and presents an upper bound on the norm of differences between SAM gradient and SGD gradient for $F$.

**Lemma L.3.** *For every $x$ satisfying $\|x\|_2 = 1$, and for every time step $t \geq 1$, the following are satisfied with high probability over the random initialization.*
*(a) For at most $\widetilde{O}(\varepsilon_a\rho\sqrt{km})$ fraction of $i \in [m]$: we have*

$$\mathbb{1}[\langle w_i^{(t+1/2)}, x\rangle + b_i^{(0)} \geq 0] \neq \mathbb{1}[\langle w_i^{(t)}, x\rangle + b_i^{(0)} \geq 0].$$

*(b) For every $r \in [k]$,*

$$\left|f_r(x;W^{(0)}+W_{t+1/2}) - f_r(x;W^{(0)}+W_t)\right| = \widetilde{O}(\varepsilon_a^3 k\rho^2 m^{3/2} + \varepsilon_a^2\sqrt{k}\rho m)$$

*(c) For every $y \in \mathcal{Y}$,*

$$\left\|\frac{\partial}{\partial W}L(F(x;W^{(0)}+W_{t+1/2}),y) - \frac{\partial}{\partial W}L(F(x;W^{(0)}+W_t),y)\right\|_{2,1}$$
$$\leq \widetilde{O}(\varepsilon_a^2 k\rho m^{3/2} + \varepsilon_a^4 k^2\rho^2 m^{5/2} + \varepsilon_a^3 k^{3/2}\rho m^2) \tag{26}$$

*Proof.* Recall that the following hold from the definition of $F$ (see Lemma B.3 of Allen-Zhu et al. (2019) for the details).

$$\left\|\frac{\partial}{\partial w_i} f_r(x; W^{(0)} + W_t)\right\|_2 \le \varepsilon_a B \quad \text{and} \quad \left\|\frac{\partial}{\partial w_i} L(F(x; W^{(0)} + W_t), y)\right\|_2 \le \sqrt{k}\varepsilon_a B \quad (27)$$

(a) Let $\tau = \varepsilon_a \rho$ and define $\mathcal{H} \overset{\text{def}}{=} \left\{i \in [m] \middle\| \left|\langle w_i^{(t)}, x\rangle + b_i^{(0)}\right| \ge 2\sqrt{k}B\tau\right\}$. Then, the lemma is a direct corollary from Lemma B.3 (a) of Allen-Zhu et al. (2019).

(b) We divide $i$ into two cases. First, when $i \notin \mathcal{H}$, we can directly utilize Lemma B.3.(b) of Allen-Zhu et al. (2019) and the total difference from these $i$'s is $\widetilde{O}(\varepsilon_a^3 k\rho^2 m^{3/2})$. Next, we consider the differences from $i \in \mathcal{H}$.

$$\left|a_{r,i}^{(0)}\left(\langle w_i^{(t+1/2)}, x\rangle + b_i^{(0)}\right)\mathbb{1}\left[\langle w_i^{(t+1/2)}, x\rangle + b_i^{(0)} \ge 0\right]\right.$$
$$\left.-a_{r,i}^{(0)}\left(\langle w_i^{(t)}, x\rangle + b_i^{(0)}\right)\mathbb{1}\left[\langle w_i^{(t)}, x\rangle + b_i^{(0)} \ge 0\right]\right|$$
$$\le \left|a_{r,i}^{(0)}\left(\langle w_i^{(t+1/2)} - w_i^{(t)}, x\rangle\right)\right|$$
$$= \left|a_{r,i}^{(0)}\left(\langle \rho \cdot \frac{\partial}{\partial w_i} L(F(x; W^{(0)} + W_t), y), x\rangle\right)\right|$$
$$\le \rho \left|a_{r,i}^{(0)}\right| \cdot \left\|\frac{\partial}{\partial w_i} L(F(x; W^{(0)} + W_t), y)\right\|_2 \cdot \|x\|_2$$
$$\le \rho(\varepsilon_a B) \cdot (\sqrt{k}\varepsilon_a B)$$
$$= \widetilde{O}(\varepsilon_a^2 \sqrt{k}\rho)$$

The first inequality is from the fact that $i \in \mathcal{H}$ and thus $\mathbb{1}\left[\langle w_i^{(t+1/2)}, x\rangle + b_i^{(0)} \ge 0\right] = \mathbb{1}\left[\langle w_i^{(t)}, x\rangle + b_i^{(0)} \ge 0\right]$. Then, we have utilized the definition of SAM (22) and Cauchy-Schwartz inequality. Since there can be at most $m$ number of $i \in \mathcal{H}$, the total differences from $i \in \mathcal{H}$ amount to $\widetilde{O}(\varepsilon_a^2 \sqrt{k}\rho m)$. Combining the two cases proves the (b).

(c) By the chain rule, we have

$$\frac{\partial}{\partial w_i} L(F(x; W^{(0)} + W_t), y) = \nabla L(F(x; W^{(0)} + W_t), y)\frac{\partial}{\partial w_i} F(x; W^{(0)} + W_t).$$

Since $L$ is 1-smooth, applying the above lemma (b) gives

$$\left\|\nabla L(F(x; W^{(0)} + W_{t+1/2}), y) - \nabla L(F(x; W^{(0)} + W_t), y)\right\|_2$$
$$\le \left\|F(x; W^{(0)} + W_{t+1/2}) - F(x; W^{(0)} + W_t)\right\|_2$$
$$\le \widetilde{O}\left(\varepsilon_a^3 k^{3/2}\rho^2 m^{3/2} + \varepsilon_a^2 k\rho m\right). \quad (28)$$

For $i \in \mathcal{H}$, we have $\mathbb{1}[\langle w_i^{(t+1/2)}, x\rangle + b_i^{(0)} \ge 0] = \mathbb{1}[\langle w_i^{(t)}, x\rangle + b_i^{(0)} \ge 0]$ and thus $\frac{\partial}{\partial w_i} F(x; W^{(0)} + W_{t+1/2}) = \frac{\partial}{\partial w_i} F(x; W^{(0)} + W_t)$. Then, combining (28) with (27) and using the fact that there can be at most $m$ number of $i \in \mathcal{H}$, this amounts to $\widetilde{O}\left(\varepsilon_a^4 k^2\rho^2 m^{5/2} + \varepsilon_a^3 k^{3/2}\rho m^2\right)$.

Next, for $i \notin \mathcal{H}$, we can directly use the result from Lemma B.3.(c) of Allen-Zhu et al. (2019) and this contributes to $\widetilde{O}(\varepsilon_a^2 k\rho m^{3/2})$. Summing these together, we prove the bound. $\qquad\square$

Finally, we show that the following lemma holds, which is a SAM version of Lemma B.4 in Allen-Zhu et al. (2019). Combined with the algorithm-independent parts presented in Allen-Zhu et al. (2019), proving the following lemma concludes the proof of Theorem L.1. We use the notation of $L_F(\mathcal{Z}; W)$ for $L_F(\mathcal{Z}; W) \overset{\text{def}}{=} \frac{1}{|\mathcal{Z}|}\sum_{(x,y)\in\mathcal{Z}} L(F(x; W + W^{(0)}), y)$ and similarly define $L_G(\mathcal{Z}; W)$.

**Lemma L.4.** *(SAM version of Lemma B.4 in* Allen-Zhu et al. (2019)*) For every* $\varepsilon \in \left(0, \frac{1}{pk\mathfrak{C}_s(\phi,1)}\right)$, *letting* $\varepsilon_a = \varepsilon/\widetilde{\Theta}(1)$, $\eta = \widetilde{\Theta}(\frac{1}{\varepsilon km})$, *and* $\rho = \widetilde{\Theta}(\frac{1}{\varepsilon^3 km^3})$, *there exists* $M = \mathsf{poly}(\mathfrak{C}_\epsilon(\phi,1), 1/\varepsilon)$ *and* $T = \Theta\left(\frac{k^3 p^2 \cdot \mathfrak{C}_s(\phi,1)^2}{\varepsilon^2}\right)$ *such that if* $m \geq M$, *the following holds with high probability over random initialization.*

$$\frac{1}{T} \sum_{t=0}^{T-1} L_F(\mathcal{Z}, W_t) \leq \mathsf{OPT} + \varepsilon. \tag{29}$$

*Proof.* Let $W^*$ be the weights constructed from the Corollary B.2 in Allen-Zhu et al. (2019). By the convexity of $L$ and Cauchy-Schwartz inequality, we have

$$
\begin{aligned}
L_G(\mathcal{Z}, W_t) - L_G(\mathcal{Z}; W^*) &\leq \langle \nabla L_G(\mathcal{Z}; W_t), W_t - W^* \rangle \\
&= \langle \nabla L_G(\mathcal{Z}; W_t) - \nabla L_F(\mathcal{Z}; W_t), W_t - W^* \rangle \\
&\quad + \langle \nabla L_F(\mathcal{Z}; W_t) - \nabla L_F(\mathcal{Z}; W_{t+1/2}), W_t - W^* \rangle \\
&\quad + \langle \nabla L_F(\mathcal{Z}; W_{t+1/2}), W_t - W^* \rangle \\
&\leq \|\nabla L_G(\mathcal{Z}; W_t) - \nabla L_F(\mathcal{Z}; W_t)\|_{2,1} \|W_t - W^*\|_{2,\infty} \\
&\quad + \|\nabla L_F(\mathcal{Z}; W_t) - \nabla L_F(\mathcal{Z}; W_{t+1/2})\|_{2,1} \|W_t - W^*\|_{2,\infty} \\
&\quad + \langle \nabla L_F(\mathcal{Z}; W_{t+1/2}), W_t - W^* \rangle
\end{aligned}
$$

From the SAM update rule (22), we have the following equality.

$$
\begin{aligned}
\|W_{t+1} - W^*\|_F^2 &= \|W_t - \eta \nabla L_F(z^{(t)}, W_{t+1/2}) - W^*\|_F^2 \\
&= \|W_t - W^*\|_F^2 - 2\eta \langle \nabla L_F(z^{(t)}, W_{t+1/2}), W_t - W^* \rangle + \eta^2 \|\nabla L_F(z^{(t)}, W_{t+1/2})\|_F^2.
\end{aligned}
$$

Thus, we have

$$
\begin{aligned}
L_G(\mathcal{Z}; W_t) - L_G(\mathcal{Z}; W^*) &\leq \underbrace{\|\nabla L_G(\mathcal{Z}; W_t) - \nabla L_F(\mathcal{Z}:W_t)\|_{2,1} \|W_t - W^*\|_{2,\infty}}_{(A)} \\
&\quad + \underbrace{\|\nabla L_F(\mathcal{Z}; W_t) - \nabla L_F(\mathcal{Z}; W_{t+1/2})\|_{2,1} \|W_t - W^*\|_{2,\infty}}_{(B)} \\
&\quad + \frac{\|W_t - W^*\|_F^2 - \mathbb{E}_{z^{(t)}}[\|W_{t+1} - W^*\|_F^2]}{2\eta} \\
&\quad + \underbrace{\frac{\eta}{2} \|\nabla L_F(W_{t+1/2}, z^{(t)})\|_F^2}_{(C)}.
\end{aligned}
$$

Since $\|W_t - W^*\|_{2,\infty} = \widetilde{O}(\sqrt{k}\varepsilon_a(\eta+\rho)t + \frac{kpC_0}{\varepsilon_a m})$, $(A)$ is bounded as

$$(A) = \widetilde{O}\left(\sqrt{k}\varepsilon_a(\eta+\rho)T\Delta + \frac{kpC_0}{\varepsilon_a m}\Delta\right)$$

where $\Delta = \widetilde{O}\left(\varepsilon_a^2 k(\eta+\rho)Tm^{3/2} + \varepsilon_a^4 k^2(\eta+\rho)^2 T^2 m^{5/2}\right)$.

Next, we can bound $(B)$ from Lemma L.3(c) as follows.

$$(B) = \widetilde{O}(\sqrt{k}\varepsilon_a(\eta+\rho)T\Delta' + \frac{kpC_0}{\varepsilon_a m}\Delta'),$$

where $\|\nabla L_F(\mathcal{Z}; W_t) - \nabla L_F(\mathcal{Z}; W_{t+1/2})\|_{2,1} \leq \Delta' = \varepsilon_a^2 k\rho m^{3/2} + \varepsilon_a^4 k^2 \rho^2 m^{5/2} + \varepsilon_a^3 k^{3/2} \rho m^2$.

We also have

$$(C) = \widetilde{O}(\eta \varepsilon_a^2 km)$$

since the norm of $\nabla L_F$ is always bounded as $\|\nabla L_F(\cdot, z^{(t)})\|_F^2 = \widetilde{O}(\varepsilon_a^2 km)$.

Then, by telescoping, we have

$$\frac{1}{T}\sum_{t=0}^{T-1}\mathbb{E}_{\text{SAM}}[L_G(\mathcal{Z};W_t)] - L_G(\mathcal{Z};W^*) \leq \widetilde{O}\left(\sqrt{k}\varepsilon_a(\eta+\rho)T\Delta + \frac{kpC_0}{\varepsilon_a m}\Delta\right)$$

$$+ \widetilde{O}\left(\sqrt{k}\varepsilon_a(\eta+\rho)T\Delta' + \frac{kpC_0}{\varepsilon_a m}\Delta'\right)$$

$$+ \underbrace{\frac{\|W_0 - W^*\|_F^2}{2\eta T}}_{(D)} + \widetilde{O}(\eta\varepsilon_a^2 km).$$

We can bound $(D)$ in the same way as Allen-Zhu et al. (2019),

$$(D) = \frac{\|W_0 - W^*\|_F^2}{2\eta T} = \widetilde{O}\left(\frac{k^2 p^2 \mathfrak{C}_{\mathfrak{s}}(\phi,1)^2}{\varepsilon_a^2 m} \cdot \frac{1}{\eta T}\right).$$

By setting $\eta = \widetilde{\Theta}(\frac{\varepsilon}{km\varepsilon_a^2}), \rho = \widetilde{\Theta}(\frac{\varepsilon}{km^3\varepsilon_a^4}), T = \widetilde{\Theta}(k^3 p^2 \mathfrak{C}_{\mathfrak{s}}(\phi,1)^2/\varepsilon^2)$, we have $\Delta = \widetilde{O}(\frac{k^6 p^4 \mathfrak{C}_{\mathfrak{s}}(\phi,1)^4}{m^{3/2}\varepsilon^4})$ and $\Delta' = \widetilde{O}(\frac{1}{m^{3/2}\varepsilon} + \frac{\sqrt{k}}{m})$. Hence, with large enough $m$, we obtain the following inequality and prove Lemma L.4, combined with the remaining parts from Allen-Zhu et al. (2019).

$$\frac{1}{T}\sum_{t=0}^{T-1}\mathbb{E}_{\text{SAM}}[L_G(\mathcal{Z};W_t)] - L_G(\mathcal{Z};W^*) \leq O(\varepsilon). \tag{30}$$

$\square$

