# OpenReview forum: "Critical Influence of Overparameterization on Sharpness-aware Minimization"
_ICLR.cc/2025/Conference — Submitted to ICLR 2025_

### Official Review · Reviewer_L6ky · 2024-10-29

**Soundness:** 3
**Presentation:** 3
**Contribution:** 3
**Rating:** 6
**Confidence:** 3

**Summary:**

This work attempts to disclose the critical influence of overparamatrisation on  Sharpness-Aware Minimization  (SAM) from experimental and theoretical perspectives. It starts with an extensive evaluation to display a  highly consistent view showing the generalization benefit of SAM increases with overparametrisation. Without such overparametrisation  SAM may not work. This leads authors to consider the benefit in terms of more solutions and implicit bias. Finally, they developed theoretical advantages and advantages of overparametrization for SAM on linear stability, convergence and generalization.

**Strengths:**

Nice empirical and theoretical work on overparametrisation on sharpness-aware minimization. It is interesting and useful for ICLR communauty

**Weaknesses:**

See questions below.

**Questions:**

(1) Please clarify the large curve surrounding the "real" curve in Fig . 1 (and possibly in other Figs). Specify if the shaded regions represent confidence intervals, standard deviations or soe other measures of uncertainty.
(2) How can SAM improvements be compared to SGD improvements via overparametrisation? Hoa SAM and SGD performance changes with increasing model size (may be you can add provide (in Annex) plots showing relative improvement over a baseline for both optimizers across different nodel sizes)
(3) Min Max optimization has some limitations (High complexity; Oscillatory behavior, sensitivity to initialization, etc.), how these problems are addressed here ? Can equation (3) (or some of its properties) help mitigate these issues ? Did you observed these specific issues in your experiments and if so, how have you addressed them ?
(4) Can we use Elliptic Loss Regularisation for SAM instead of Min Max? Please comment on whether Elliptic Loss Regularization could be a viable alternative to the min-max formulation in SAM, and what potential advantages or disadvantages it might have in this context.
(5) Is there a sort of "optimal" number of parameters for which the computation price is acceptable ?Did you  observed any diminishing returns in performance improvement as model size increased. Please provide guidance on balancing computational cost with performance gains when using SAM in practice

---

> ### Author Response · Authors · 2024-11-24
> **Response to L6ky**
>
> We sincerely thank the reviewer for finding our work interesting and useful. We also appreciate the reviewer’s constructive feedback. While we make clarifications to specific comments below, please let us know if there is anything we need to address further. We would be keen to engage in any further discussion.
>
> ---
>
> **(1) Meaning of shaded region**
>
> We use standard error (ie, $se = std / \sqrt{n}$, where $n=3$ for three different random seeds), with the shaded region representing $\text{avg} \pm \text{se}$.
>
> ---
>
> **(2) SAM improvements vs. SGD improvements**
>
> We did not fully grasp exactly what the reviewer is requesting. For relative improvement of SAM compared to SGD, we refer to Figure 1. If it is for the individual improvement of SAM and SGD over overparameterization, the validation accuracy of each optimizer is presented in Figure 7, Appendix B. If none of these align with your request, we would greatly appreciate further clarification.
>
> ---
>
> **(3) Min Max issue of SAM?**
>
> We have not observed these issues related to min-max optimization in our experiments. We also have not found any reports of such in prior SAM-related research either.
>
> We hypothesize that this is because **the inner maximization problem of SAM in Equation (2) is constrained within a small norm-ball**, which could potentially mitigate the typical instabilities associated with min-max optimization.
>
> Having said that, there are some prior studies worth noting regarding the stability of SAM. For instance, one study [1] suggests that SAM may be slightly more prone to convergence towards saddle points compared to SGD. On the other hand, another study [2] shows that SAM is less sensitive to the choice of activation functions compared to similar methods, such as those incorporating gradient penalties. Additionally, while not directly related to SAM (as its inner maximization is solved in closed form at each step, unlike GDA), recent studies on GDA-based solvers for min-max problems [3] might be of interest.
>
> Recognizing its worth, we will include this discussion in the revised version.
>
> ---
>
> **(4) Can Elliptic Loss Regularization replace SAM?**
>
> We appreciate the reviewer for the interesting question. We are unable to predict for certain whether ELR could be a viable alternative, however, because even for well-studied classic sharpness-minimizing algorithms (e.g., gradient norm penalties, white noise), ongoing research [2, 4] continues to uncover their hidden differences. Given that ELR has yet to be analyzed in similar depth, it is difficult to ascertain precisely how it would compare to SAM.
>
> Nonetheless, expanding our analysis on SAM to ELR could provide insight into this question, which could be an interesting direction for future research as mentioned in “Other sharpness minimization schemes”, Section 7.
>
> ---
>
> **(5) Optimal number of parameters?**
>
> > Is there a sort of "optimal" number of parameters for which the computation price is acceptable ?Did you observed any diminishing returns in performance improvement as model size increased. Please provide guidance on balancing computational cost with performance gains when using SAM in practice
>
> This is a great question indeed. Although we did not find a clear sign of diminishing returns in our experiments, it is admissible that the performance is not expected to keep on improving linearly with increasing parameters, indefinitely. The ground for this hypothesis is based on our ablation results in Section 5 where we find that the benefit of overparameterization is not always obtained when there is insufficient regularization, and from which, we may project that only scaling models (without concerning data for instance) can fall into an insufficient regularized case.
>
> Nevertheless, we conjecture that if one is given “enough” resources (in terms of computations, memory, and data), we are still optimistic that the “optimal” number of parameters may be observed to be very large. While this is rather impossible to confirm, and it is therefore challenging for us to provide a solid "guidance", we can surely measure improvements vs. parameter increase, and further, extend our discussion in this aspect.
>
> Once again, we are grateful for the reviewer’s very insightful suggestion.
>
>
> &nbsp;
>
> Reference \
> [1] Kim et al., Stability Analysis of Sharpness-Aware Minimization. arXiv, 2023. \
> [2] Dauphin et al., Neglected Hessian component explains mysteries in sharpness regularization. NeurIPS, 2024. \
> [3] Kaddour et al., When Do Flat Minima Optimizers Work? NeurIPS, 2022.

---

> > ### Comment · Reviewer_L6ky · 2024-11-26
> >
> > I thank the authors for responding to my comments/concerns, I read their rebuttal as well as the other reviews. For now, I remain on the side of acceptance with the same score. Thanks!

---

> > > ### Author Response · Authors · 2024-11-27
> > > **Thank you**
> > >
> > > We sincerely thank the reviewer for their positive comments and constructive feedback. Please let us know if there is anything we can address further.

---

### Official Review · Reviewer_8b2s · 2024-10-31

**Soundness:** 2
**Presentation:** 3
**Contribution:** 1
**Rating:** 6
**Confidence:** 2

**Summary:**

This paper claimed that SAM benefits from the over-parameterization. They find that SAM could find different (e.g. simpler) solutions than SGD in the over-parameterization regime. In addition, they find that SAM is robust to label noise and sparse parameterization. They also include some theoretical analyses, thought these analyses are irrelevant to their main findings.

**Strengths:**

1. The authors picked up an interesting perspective to investigate the mechanism of SAM. This research topic is significant.
2. The experiments are extensive, across various datasets and architectures.
3. The writing structure is clear, and the paper is easy to follow.

**Weaknesses:**

Firstly, for the empirical findings,

1. It is unclear that whether over-parameterization consistently helps SAM find more generalizable solutions as the model parameters scale up. As shown in Fig.1 and Fig. 7 (I think the Fig. 7 is clearer), for MLP on MNIST, ResNet-18 on CIFAR-10, LSTM on SST2 and CNN on Atari, when model get more over-parameterized, the generalization improvement does not always increase; sometimes, a noticeable decline is observed. Moreover, as noted by the authors, when there is no sufficient regularization, SAM might not benefit from the over-parameterization (as shown in Fig. 5). In addition, I notice that the authors conducted experiments for ViT on CIFAR-10 in Fig.5 but not in Fig. 1. I wish the authors could include the results for ViT on CIFAR-10 in Fig.1.
2. Increasing depth of models yields different conclusions than increasing width. In Fig. 1, authors increase the number of neurons in each layer, leading to a seemingly consistent conclusion that over-parameterization helps SAM find generalizable solutions. However, in Fig. 17, when increasing the number of layers, SAM might not benefit from the over-parameterization (ResNet on CIFAR-10). Clearly, the increase of both depth and width demonstrates the over-parameterization, however, the conclusion differs.
3. It is not surprising that the marginal improvement of SAM over SGD increases with higher label noise. In Fig. 12, in the over-parameterization regime, the performance of SGD drops significantly with higher label noise rate; however, SAM remains good generalization ability. Indeed, it is well known that SAM is robust to label noise, while SGD not. Thus, it is not surprising for us to observe such phenomenon. Also, it seems that the authors over-claimed that over-parameterization secures the robustness of SAM to label noise. Indeed, the observed increase of margin improvement of SAM over SGD with larger model parameters is primarily due to the pronounced decline in SGD's performance as the model parameters scale up under high label noise conditions.
4. The conclusion on the effect of sparsity contradicts with the main argument that SAM benefits from the over-parameterization. In Fig. 5, it is observed that the generalization improvement tends to increase as the model becomes sparser. However, increased sparsity indicates that the model is becoming more under-parameterized. The authors are suggested to explain such contradiction.
5. In Fig. 2 and 3, the authors compare GD and SAM, however, a comparison between SGD and SAM is more preferred.
6. In section 4.2, “implicit bias of SAM increase with over-parameterization” is over-stated. In this section, the authors demonstrated that we could use larger perturbation radius when scaling up the model parameters, however, this doesn’t indicate that a larger scale of model parameters induces stronger implicit bias towards flatter minima.

In short, SAM doesn’t always benefit from the over-parameterization in practice. A more thorough justification is needed to clarify whether over-parameterization is advantageous for SAM and under what specific conditions it may provide benefits. Indeed, it is quite expectable that SAM would benefit from over-parameterization sometimes. Because in the under-parameterization regime, the model capacity is low (say only one solution in the solution space), thus both SGD and SAM achieve similar solutions. Once scaling up the parameters, the solution space is getting larger, thereby SAM could differ from SGD and find more generalizable solutions. However, SAM might not always from the over-parameterization.

Second, for the theoretical analysis,

1. As noted by the authors, the theoretical analysis cannot support the findings in Section 3 and 4, and thus the significance of the theoretical analysis is limited. The section 6 is more like a collection of possible theoretical analyses that could be done for SAM.
2. Indeed, there is no clear theoretical support for the main discovery of this paper, as noted by the authors in the limitations part in Section 7. This is another critical issue of this paper.

Overall, the major issue of this paper is the novelty. The empirical findings are largely anticipated, and the theoretical analysis closely follows prior works without introducing any new proof techniques.  Also, some statements are clearly over-claimed. The contribution of this paper is marginal, and thus I lean to rejection.


----------------
As I cannot respond in official comments further, I put my responses here:

Based on your clarification on the ViT experiment, I decide to increase my score from 5 to 6. However, I strongly recommend the authors to revise the paper from the following aspects:

- **Overstated claims**: Some statements, like "over-parameterization consistently improves the generalization benefit of SAM" in the introduction part, are over-stated, which might be misleading for the readers. Some failure experiments such as ViT, increasing depth of networks, are encouraged to put earlier in the main text, along with throughout discussions. *I firmly believe that adhering to strictness is not a bad thing for a scientific paper*.
- **Experiments on Absolute validation metrics**: Experiments using the absolute validation metrics are encouraged to put into main text along with the experiments using "Acc improvements".
- **Current theory cannot directly support your main finding**: Still, currently, your theory cannot directly support your finding. I am not diminishing your contribution, but clearly those complementary theories in current paper cannot persuade me to highly evaluate your paper. I wish the authors could consider my suggestions and improve the paper with more related theoretical analyses.

BTW, I decide to keep my confidence score low.

**Questions:**

N/A

---

> ### Author Response · Authors · 2024-11-17
> **Response to 8b2s (1/2)**
>
> We thank the reviewer for the detailed review. While we partly relate to the reviewer’s concerns, we believe the criticism is unfairly harsh.
>
> The reviewer primarily raises two issues:
> - The empirical findings are deemed “anticipated,” and
> - The theoretical analyses are considered irrelevant to the “main findings.”
>
> We believe these criticisms are often:
> - subjective, if not biased (e.g., “contributions are marginal”),
> - groundless (e.g., “empirical findings are anticipated”),
> - based on misunderstandings (e.g., “theoretical analysis introduce no new proof techniques”),
> - and underappreciating our contributions.
>
> Below is our rationale.
>
> First, we think the reviewer’s assertion that the empirical results are “anticipated” is rather superficial and dismissive for the following reasons.
> - There were uncertainties that made such confident predictions challenging (e.g., ensuring consistency across all diverse domains, novel observations on $\rho$, sparsity, inefficient regularizations that lack any established point of reference).
> - Some of the reviewer’s comments are clearly based on a misunderstanding (e.g. “The conclusion on the effect of sparsity contradicts with the main argument”).
> - There is a significant difference between anticipating something from intuition and explicitly verifying it through extensive experimentation (e.g., “it is quite expectable that SAM would benefit from over-parameterization sometimes”). If there are specific references the reviewer is drawing upon for this anticipation, we kindly ask the reviewer to share them, and we will review them attentively.
>
> Also, regarding the contributions of our theoretical analyses, we make the following points.
> - There are novel results on higher-order Hessian moments in Section 6.1 that are not found in prior linear stability analysis of SGD.
> - There are non-trivial proof techniques newly introduced in Section 6.2 that are overlooked. For instance, we developed Lemma I.1 and I.2 to utilize the similarity between SAM and SGD gradients. This precise formulation has not been introduced in any prior work to our knowledge.
> - Our work may not offer a pinpoint theoretical explanation for Section 3. However, doing so is a significant challenge, since there is no sufficient prior theoretical analysis on modeling and comparing the generalization of two different optimizers under varying degrees of overparameterization. If the reviewer finds this task trivial, we would really appreciate any specific guidance on how it can be accomplished. Otherwise, we believe it is unfair to reject a study simply based on the absence of theoretical/mathematical explanations for its empirical discoveries, without considering the inherent challenges posed by the current state of the literature.
> - Nonetheless, it should be noted that we have presented reasonable explanations for the account of the empirical findings in Section 4. Please point us to know if there is any fundamental issue with our explanation.
>
> Finally, the reviewer’s claim that some of our conclusions are over-claiming seems to stem from either overlooking essential elements of our paper (**W1-6**) or based on subjective opinion (**W1-3**).
>
> In short, we believe the reviewer’s opinions are often subjective and unnecessarily harsh. Nonetheless, we respond to the reviewer’s specific comments as below.
>
> ---
>
> **W1-1. Consistency of the benefit**
>
> > the generalization improvement does not always increase; sometimes, a noticeable decline is observed.
>
> While results are not strictly monotonic at all intermediate points, the overall trend in generalization improvement is clear and consistent across a wide variety of workloads. We find this criticism unnecessarily castigating, focusing too narrowly on minor deviations while overlooking the broader trend rigorously validated through an extensive amount of training runs.
>
> > Moreover, as noted by the authors, when there is no sufficient regularization, SAM might not benefit from the over-parameterization (as shown in Fig. 5).  I wish the authors could include the results for ViT on CIFAR-10 in Fig.1.
>
> The aim of Figure 5 is to show that sufficient regularization is needed to benefit from overparameterization, with ViT trained on CIFAR-10 serving as an example with limited inductive bias compared to CNNs [1]. Typically, ViTs are supposed to be pretrained beforehand on datasets significantly larger than ImageNet, but then it will not align with the setup of Figure 1 where models are trained from scratch. Therefore, it is not suitable to move the results for ViT to Figure 1.

---

> ### Author Response · Authors · 2024-11-17
> **Response to 8b2s (2/2)**
>
> **W1-2. Impact of scaling depth**
>
> > Increasing depth of models yields different conclusions than increasing width. ... in Fig. 17, when increasing the number of layers, SAM might not benefit from the over-parameterization (ResNet on CIFAR-10).
>
> We agree that increasing depth in ResNet shows a less significant effect than increasing the width. We note that there are more complicated factors in increasing depths such as deciding whether to increase the number of resblocks, layers within the resblock, or all of these within each stage having different width ratios, which could influence the training dynamics in different ways. More importantly, however, we do not intend to argue that overparameterization always benefits SAM, and the results of Figure 17 seem to indicate the influence of these architectural factors. We will clarify this in the revised version.
>
> ---
>
> **W1-3. Conclusion for label noise is an over-claim**
>
> > It is not surprising that the marginal improvement of SAM over SGD increases with higher label noise.
>
> We regret that the reviewer seems to be misunderstanding the focus of this experiment, which is observing the behavior under various levels of overparameterization, rather than the degrees of label noise. Figure 12 (a-d) clearly shows that the generalization improvement of SAM increases with more parameters regardless of label noise ratio. While we are aware that SAM’s robustness to label noise is well known, our core message here is not about “the marginal improvement of SAM over SGD increases with higher label noise”, but rather that the benefit of SAM under label noise becomes more pronounced with overparameterization and less apparent with fewer parameters.
>
> > Also, it seems that the authors over-claimed that over-parameterization secures the robustness of SAM to label noise.
>
> With respect, we find the reviewer’s opinion that we are over-claiming to be quite subjective or at least based on misunderstanding. As demonstrated in Figure 12 (b-d), we observe that not only does the gap between SAM and SGD increase with more parameters, but the validation accuracy of SAM also increases while that of SGD decreases. Also, less parameter leads to worse validation accuracy, even in high label noise. This indicates that SAM itself is increasingly robust with more parameter count under label noise. As far as we know, this result has not been previously reported, and we consider the evidence sufficient to support our statement. While we demonstrated this only through the gap with the non-sharpness-minimizing baseline (i.e., SGD) in the main text, we will further clarify this in the revised version.
>
> ---
>
> **W1-4. Sparsity experiments contradicting the main message?**
>
> We regret that the reviewer is misunderstanding the setup of the experiment regarding sparsity. As already mentioned in Lines 308-309, the models do not get underparameterized with increasing sparsity since we increase the width of the model proportional to the increasing sparsity such that the number of parameters matches the original dense model.
>
> ---
>
> **W1-5. SGD instead of GD in Figures 2 and 3?**
>
> > In Fig. 2 and 3, the authors compare GD and SAM, however, a comparison between SGD and SAM is more preferred.
>
> We used GD over SGD to better compare the effect of SAM on the solution without the effect of mini-batch stochasticity following [2]. Also, we found the result for SGD vs. (stochastic) SAM to yield similar results.
>
> ---
>
> **W1-6. Section 4.2 is over-stating?**
>
> We believe that our argument in Section 4.2 is reasonable, and the reviewer’s concerns seem to stem from overlooking key elements of our argument. Specifically, we reference a prior study [3] in Equation (4), which demonstrates how a larger value of $\rho$ leads to stronger implicit regularization. Also, Equation (6) shows that an increase in $\rho$ results in a tighter upper bound on the maximum Hessian eigenvalue, thus biasing towards flatter minima. Along with our observations, these points were central to our conclusion that SAM’s implicit bias increases with overparameterization.
>
> ---
>
> **Final remark**
>
> We sincerely thank the reviewer for taking the time to review our work in such detail. However, with all due respect, we believe that the reviewer’s overall critiques are unfairly harsh and often subjective. We sincerely hope the reviewer reads our responses and reconsiders the value of our work. If there are any concerns you would like to further discuss with us, please specify them so we can address them further during the discussion period.
>
> &nbsp;
>
> **References** \
> [1] Dosovitskiy et al., An Image is Worth 16x16 Words: Transformers for Image Recognition at Scale. ICLR, 2021. \
> [2] Andriushchenko and Flammarion. Towards understanding sharpness-aware minimization. ICML, 2022. \
> [3] Compagnoni et al., An SDE for Modeling SAM: Theory and Insights. ICML, 2023.

---

> ### Comment · Reviewer_8b2s · 2024-11-23
>
> Thank you for the detailed responses. First of all, I would like to clarify that **all the concerns raised in my review grounded in evidence presented within your paper**. I believe my concerns are neither “subjective” nor “groundless”.  If you disagree with my assessment, I welcome further discussion; however, I kindly request that the focus remains on addressing the points to improve the overall quality of the paper.
>
> > There are novel results on higher-order Hessian moments in Section 6.1 that are not found in prior linear stability analysis of SGD.
>
> Technically speaking, in Definition H.1, the Hessian moments come from the linearized update rule of SAM and the proof is indeed similar to Wu et al. (2018).
>
> > Our work may not offer a pinpoint theoretical explanation for Section 3. However, doing so is a significant challenge, since there is no sufficient prior theoretical analysis on modeling and comparing the generalization of two different optimizers under varying degrees of overparameterization. If the reviewer finds this task trivial, we would really appreciate any specific guidance on how it can be accomplished. Otherwise, we believe it is unfair to reject a study simply based on the absence of theoretical/mathematical explanations for its empirical discoveries, without considering the inherent challenges posed by the current state of the literature.
>
> First, my concern here is that “there is no clear theoretical **support** for the main discovery of this paper”. I am **not** requesting you to develop new theory but concerning about the logical flow. In my view, **an acceptable paper** should present its logical reasoning clearly and provide sufficient justification for its findings. If your work uncovers something surprising, it is essential to accompany it with an explanation—either through carefully designed ablation studies or robust theoretical support. However, the theoretical analysis in Section 6 does not seem to address or explain the main discovery presented in Section 3. This **disconnect** significantly diminishes the impact and value of the work. I encourage the authors to bridge this gap to strengthen the paper’s overall contribution.
>
> > There are non-trivial proof techniques newly introduced in Section 6.2 that are overlooked. For instance, we developed Lemma I.1 and I.2 to utilize the similarity between SAM and SGD gradients. This precise formulation has not been introduced in any prior work to our knowledge.
>
> Still, my concern is not solely on the techniques but the logical flow of your paper. The findings in Sec. 3 are requested to be carefully justified.
>
> ----------------------------
>
> > While results are not strictly monotonic at all intermediate points, the overall trend in generalization improvement is clear and consistent across a wide variety of workloads.
>
> As noted, therefore, in line 065, “over-parameterization consistently improves the generalization benefit of SAM” is **overstated**.
>
> > Typically, ViTs are supposed to be pretrained beforehand on datasets significantly larger than ImageNet, but then it will not align with the setup of Figure 1 where models are trained from scratch.
>
> First, training ViTs on small datasets are possible and common practice now [1, 2]. Then, how did you train ViTs in Fig. 5? Did you finetune it from some pretrained checkpoint on CIFAR-10? If so, it is unfair to compare finetuning with training from scratch. I wish the author could give more details about the experiments in Fig. 5.
>
> [1] Hanan Gani, Muzammal Naseer, Mohammad Yaqub. How to Train Vision Transformer on Small-scale Datasets? In BMVC, 2022.
>
> [2] https://github.com/lucidrains/vit-pytorch
>
> > We agree that increasing depth in ResNet shows a less significant effect than increasing the width.
>
> Then, some arguments in the main section are overstated. Also, there is **no direct reference to Fig. 17** in the main text. I hope the authors could mention their investigation of model depth in the main text in their revision.
>
> > Figure 12 (a-d) clearly shows that the generalization improvement of SAM increases with more parameters regardless of label noise ratio.
>
> Thank you for directing me to Fig. 12. Now I clearly see the benefits of SAM in the present of label noise.
>
> > Also, we found the result for SGD vs. (stochastic) SAM to yield similar results.
>
> Some results on SGD vs. SAM are encouraged to be included in your revision.
>
> > demonstrates how a larger value of ρ leads to stronger implicit regularization.
>
> I believe a more principal way is to measure the flatness directly.

---

> > ### Author Response · Authors · 2024-11-24
> > **Addressing additional comments (1/2)**
> >
> > > “the proof is indeed similar to Wu et al. (2018)”
> >
> > Please note, (i) the analysis from Wu et al. [1] is not for SAM, (ii) Definition H.1 (linearized stochastic SAM) is what we develop specifically for the purpose of analyzing SAM, and more importantly, (iii) our result of the upper bound for the higher-order Hessian moments in SAM, to the best of our knowledge, has not been revealed in any prior studies, which has been recognized by other reviewers.
> >
> > We also find the criticism that “the proof is similar” (and by implication, trivial) to be difficult to agree with for the following reasons: (1) bounding the update iterate is an obvious and natural step for analyzing any optimization methods, and (2) we conducted standard algebraic manipulations and spectral inequality in expanding the linearized SAM, which are common tools naturally adopted in any mathematical derivation in optimization. The reviewer seems to be suggesting that entirely new sets of mathematical tools and skills are necessary for any results to be recognized as novel, which we find unreasonable.
> >
> > ---
> >
> > > “I am not requesting you to develop new theory but concerning about the logical flow.”
> >
> > It seems that the reviewer has missed an important aspect of our work, that is, Section 4. As addressed already in our previous response, Section 4 provides theoretical and empirical accounts of the findings in Section 3, which we believe is sufficiently logical and sound in explaining why SAM may benefit from overparameterization, and also, why it may not without overparameterization. We would appreciate it if the reviewer precisely points us to exactly what is “logical flow” in our work. We would also really appreciate the reviewer if the reviewer could provide us with constructive feedback and specific suggestions regarding Section 4.
> >
> > ---
> >
> > > “In my view, an acceptable paper should present its logical reasoning clearly and provide sufficient justification for its findings. If your work uncovers something surprising, it is essential to accompany it with an explanation—either through carefully designed ablation studies or robust theoretical support.”
> >
> > Again, please point us to exactly where we failed to provide “accompanying explanation” or why Section 4 cannot serve to explain our findings in Section 3. Also, please let us know why Section 5 cannot serve as “carefully designed ablation studies”. We will try our best to reflect any concrete/specific suggestions in the revised version.

---

> ### Author Response · Authors · 2024-11-24
> **Addressing additional comments (2/2)**
>
> > “over-parameterization consistently improves the generalization benefit of SAM” is overstated.
>
> We used the term “consistent” to indicate “tendency” or “trend”, rather than “monotonic”. We believe that by reading our paper one would not be mistaken by that. We can fix this in the final version. We however leave a clear note to the reviewer: We do not disagree that this level of strictness can indeed improve the paper, and yet, we feel that this is quite a nitpicking rather than a constructive criticism, and we would strongly disagree if this constitutes one of the major reasons for rejecting the paper.
>
> ---
>
> > Did you finetune it from some pretrained checkpoint on CIFAR-10? If so, it is unfair to compare finetuning with training from scratch. I wish the author could give more details about the experiments in Fig. 5.
>
> No. We didn’t finetune it. We train it from scratch as with all other cases. So it’s not unfair.
>
> To repeat, the purpose of Section 5 is to show that overparameterization may not help SAM without sufficient regularization (this is the nuance we pitched in the paper as “caveats”). Here, ViT (without pretraining, thus no inductive bias) was chosen as an example case of insufficient regularization along with other cases of no weight decay and no early stop. We can make it clearer in the final version.
>
> ---
>
> > I hope the authors could mention their investigation of model depth in the main text in their revision.
>
> We will address this in the revision.
>
> > Some results on SGD vs. SAM are encouraged to be included in your revision.
>
> We will include this in the revision.
>
> ---
>
> > I believe a more principal way is to measure the flatness directly (insead of demonstrating through larger rho).
>
> We regret that the reviewer seems to be misunderstanding the overall purpose of Section 4. The goal of Section 4 is not to measure flatness but rather to provide a theoretical account of how SAM might benefit from overparameterization. Here, the reason we provide our observations on $\rho$ is to provide evidence supporting the theoretical result that the implicit bias (not flatness in particular) of SAM can increase with overparameterization.
>
> Additionally, we have provided $\lambda_{\text{max}}(H)$ for SGD and SAM in Figure 6-(a), which is a commonly used and widely accepted measure for sharpness. If the reviewer believes that a “more principled way” of measuring flatness is necessary, we would appreciate clarification on what specific method to adopt and an explanation of why this alternative would be preferable.

---

> ### Comment · Reviewer_8b2s · 2024-11-25
>
> Thank you for the responses. I am a little bit confused.
>
> In the your latest response:
>
> > No. We didn’t finetune it. We train it from scratch as with all other cases. So it’s not unfair.
>
> Okey. Now, the experiments in Fig. 5 trained ViTs from scratch. But in your earlier response:
>
> > Typically, ViTs are supposed to be pretrained beforehand on datasets significantly larger than ImageNet, but then it will not align with the setup of Figure 1 where models are trained from scratch. Therefore, it is not suitable to move the results for ViT to Figure 1.
>
> Then, I believe the rationale for not including the results for ViT in Figure 1 is unconvincing.
>
> BTW, I have increased my score from 3 to 5 based on the clarification of the purpose of Sec 4, and lowered my confidence because of some unresolved issues. I will come back to the other issues later. In addition, **a revision of paper is suggested** during the discussion phase.

---

> > ### Author Response · Authors · 2024-11-25
> > **Thank you**
> >
> > We sincerely appreciate the reconsideration and the increase in score. We will promptly revise the manuscript and ensure it is uploaded before the deadline. Also, further clarification regarding ViT will follow shortly. In the meantime, please feel free to share any additional concerns you would like to discuss.

---

> > > ### Author Response · Authors · 2024-11-28
> > > **Revision**
> > >
> > > We have made the following changes to the paper:
> > >
> > > - A discussion of the depth experiments has been added to Section 7, “More Ablation Study”.
> > > - Additional comparisons between the solutions and trajectory of SGD and SAM are provided in Figures 8 and 9 in Appendix G.

---

> ### Author Response · Authors · 2024-11-29
> **Response regarding ViT**
>
> > “the rationale for not including the results for ViT (on CIFAR-10) in Figure 1 is unconvincing”
>
> The reason why ViT is not included in Figure 1, despite the fact that it is trained from scratch, is that training ViT from scratch is an example case of “insufficient regularization” (due to lack of inductive bias [1, 2]) where SAM may not benefit much from overparameterization. In this work, we say that overparameterization can be helpful for SAM, but that there may be “caveats” as well, so it should be interpreted with caution (Section 5). Specifically, in Section 5, we show that sufficient regularization is necessary to obtain the benefit of overparameterization, and to demonstrate this, we present three representative examples: (a) without weight decay, (b) without early stopping, and (c) without sufficient inductive bias. Our ViT experiments correspond to (c), and this is why it was included in Figure 5 rather than Figure 1.
>
> We hope this clears the confusion about ViT experiments, but if it needs any further clarification, please let us know. We would be happy to explain further.
>
> &nbsp;
>
> **References**\
> [1] Dosovitskiy et al., An Image is Worth 16x16 Words: Transformers for Image Recognition at Scale. ICLR, 2021.\
> [2] Hanan Gani, Muzammal Naseer, Mohammad Yaqub. How to Train Vision Transformer on Small-scale Datasets? BMVC, 2022.

---

> > ### Author Response · Authors · 2024-11-29
> > **Awaiting your feedback**
> >
> > Dear reviewer,
> >
> > We have addressed the specific points in your latest response. We would greatly appreciate it if you could review them. Please let us know if there is anything else you want us to explain more. We would be happy to engage in any further discussions.
> >
> > Regards,\
> > Authors

---

> > ### Comment · Reviewer_8b2s · 2024-12-02
> >
> > Thank you for your feedback. I have some further question: where does this induction bias come from? Is it from the transformer-based architecture? Moreover, can this induction bias be mitigated by finetuning a pretrained ViT model? This is important, as transformer-based architecture is more frequently used in nowadays deep learning applications.

---

> > > ### Author Response · Authors · 2024-12-03
> > > **Additional response regarding ViT**
> > >
> > > > where does this induction bias come from? Is it from the transformer-based architecture?
> > >
> > > Yes. We refer the reviewer to the following: “The inductive bias of a network is determined by its architecture, and judicious choices of model can drastically improve generalization.” - Section 20.4.3, p412, [1]
> > >
> > > ---
> > >
> > > > Moreover, can this induction bias be mitigated by finetuning a pretrained ViT model?
> > >
> > > First of all, **inductive bias here is a positive thing** and hence, it's something that you would induce rather than "mitigate".
> > >
> > > We also inform the reviewer that it is well known that **ViT lacks inductive bias favorable for the image domain**, and thus normally accompanies large-scale pretraining [1-3]. This is nicely summarized in [3] as follows: “Despite their advantages, ViTs fail to match the performance of CNNs when trained from scratch on small-scale datasets. This is primarily due to the lack of locality, inductive biases and hierarchical structure of the representations which is commonly observed in the CNN architectures. As a result, ViTs require large-scale pre-training to learn such properties from the data for better transfer learning to downstream tasks.”
> > >
> > > We leave a clear note here that even if it were hypothetically possible to remove some inductive bias from ViTs---which we find quite arbitrary---it would still _not_ be suitable for Figure 1 because it would require a very different training procedure from other workloads presented in Figure 1. Nonetheless, we are open to providing additional experiments on this as a supplement if requested.
> > >
> > > ---
> > >
> > > > This is important, as transformer-based architecture is more frequently used in nowadays deep learning applications.
> > >
> > >
> > > With all due respect, we are quite frustrated by the reviewer raising this point. Thus far, we have aimed to clarify what seemed to be a misunderstanding regarding the purpose and configuration of the ViT experiment.
> > >
> > > However, now this latest comment appears to be only concerned with “transformer-based architecture” as if for some experiment to be valid, it has to deal with transformers or be related to deep learning applications, which completely ignores the purpose of the experiments and diminishes our contributions.
> > >
> > > While we welcome any further discussion and add the discussions we had with the reviewer in the final version, we sincerely request the reviewer to re-examine our experiments surrounding ViT experiments and re-consider our contributions.
> > >
> > > &nbsp;
> > >
> > > **References** \
> > > [1] Prince, Understanding Deep Learning. 2023. \
> > > [2] Dosovitskiy et al., An Image is Worth 16x16 Words: Transformers for Image Recognition at Scale. ICLR, 2021. \
> > > [3] Hanan Gani, Muzammal Naseer, Mohammad Yaqub. How to Train Vision Transformer on Small-scale Datasets? BMVC, 2022.

---

> > > > ### Author Response · Authors · 2024-12-04
> > > > **Closing remark**
> > > >
> > > > As the end of the discussion period is closing, we’d like to sincerely thank the reviewer for spending their time and efforts reviewing our paper and providing constructive feedback, and especially for raising the score. We will make our best efforts to better reflect your suggestions in the final version.

---

### Official Review · Reviewer_18YL · 2024-11-01

**Soundness:** 2
**Presentation:** 3
**Contribution:** 2
**Rating:** 5
**Confidence:** 4

**Summary:**

This paper studies the critical influence of overparameterization on SAM.

**Strengths:**

The experiments in this paper are insightful, showing that overparameterization can increase the performance gap between SAM and SGD, while also highlighting the role of factors such as label noise, sparsity, weight decay, and early stopping in this phenomenon.

**Weaknesses:**

- It is widely recognized that overparameterization improves the generalization performance of SGD (e.g., ResNet-152 outperforms ResNet-18 on Cifar10 using SGD). And it is unsurprising that overparameterization can also improve the generalization performance of SAM. However, the non-trivial aspect is that overparameterization increases *the gap between SAM and SGD*, as shown in Figure 1. To avoid trivialization, reconsidering the title and abstract might be beneficial.

- My primary concern is that the theoretical analysis can not support the main claim/findings (e.g., Figure 1).
  - The linear stability analysis in Section 6.1 can not demonstrate how overparameterization affects SAM (i.e., that “greater overparameterization leads SAM to find flatter minima”).
  - In the convergence analysis in Section 6.2: (i) it lacks relevance to generalization, the core focus of the paper; (ii) it does not clarify how the degree of overparameterization influences the convergence speed within the interpolation regime; (iii) in the interpolation regime, the exponential convergence result also holds for SGD (Bassily et al., 2018). The proof for SAM appears to be a minor extension from SGD, which treats SAM as SGD + small permutation.
  - The theoretical analysis in Section 6.1/6.2 focus on a variant of SAM (SAM without normalization) for simplification, rather than the original SAM. While not a major issue, the formulation should be clearly presented in Section 6 to prevent potential reader confusion.

**Questions:**

Could the authors provide adequate theoretical support for the main findings? e.g., an analysis on diagonal linear networks.

---

> ### Author Response · Authors · 2024-11-14
> **Response to 18YL (1/2)**
>
> We appreciate the reviewer for taking the time to review our work. While we respond to the reviewer’s specific comments as below, we would be keen to engage in any further discussion.
>
> ---
>
> **Theoretical support**
>
> > My primary concern is that the theoretical analysis can not support the main claim/findings (e.g., Figure 1).
>
> We clarify that the purpose of Section 6 is to provide diverse theoretical benefits that overparameterization has on SAM, rather than to support the observations in Section 3 (e.g., Figure 1) as already mentioned in Section 6 (see Line 359). We provide our accounts of these findings in Section 4, which might not be entirely “theoretical”, yet plausible and reasonable. Please let us know if there is any issue with it.
>
> On a slightly different note, we believe that Section 6 itself, independently of its connection to Section 3, is noteworthy in that the results have not been explicitly developed in the literature, rendering its unique contributions. We address specific points in detail below.
>
> ---
>
> **Section 6.1**
>
> > The linear stability analysis in Section 6.1 can not demonstrate how overparameterization affects SAM (i.e., that “greater overparameterization leads SAM to find flatter minima”).
>
> We respectfully disagree with this assertion. Equation (6) shows that a larger $\rho$ provides a tighter upper bound on the maximum Hessian eigenvalue, which leads SAM to find flatter minima. We emphasize that this property not only holds when overparameterization (i.e., interpolation) is achieved (see Line 392), but in conjunction with the findings in Section 4.2 that larger models tend to have higher optimal $\rho$ values, suggesting that greater overparameterization can guide SAM toward flatter minima.
>
> ---
>
> **Section 6.2**
>
> > (i) it lacks relevance to generalization, the core focus of the paper
>
> As clarified at the beginning, the purpose of Section 6.2 is not to address generalization properties, but to present the influence of overparameterization on the convergence of SAM. We believe these results are non-trivial in that for the first time, SAM is shown explicitly to converge at a linear convergence rate under overparameterization, which is much faster than the previously known sublinear rate, implying one critical influence of overparameterization on SAM (i.e., improved convergence).
>
> > (ii) it does not clarify how the degree of overparameterization influences the convergence speed within the interpolation regime
>
> Our finding indicates that with the existence of overparameterization, the convergence speed improves from sublinear to linear rate. To the best of our knowledge, we are unaware of any theories or analyses that capture varying degrees of overparameterization applicable to our study. We conjecture that formulating such a notion in the context of interpolation is complex, and our understanding as a society is still limited. Nonetheless, we would be keen to find any such work, so please let us know if the reviewer could suggest any. We are eager to pursue further study in this direction.
>
> > (iii) The proof for SAM appears to be a minor extension from SGD
>
> We respectfully disagree with this assessment, as nontrivial efforts were made in the proof. Specifically, for Theorem 6.6, Lemma I.1 and I.2 were developed to characterize the similarity between SAM gradient and SGD gradient without relying on any special assumptions that might be infeasible in practice. This precise characterization, to the best of our knowledge, has not been presented in any previous works. We remark that it is a standard procedure to build upon existing results in optimization, and indeed, many prior works on extending the result for SGD to other optimizers [2-3] have been well-received by the community. In this regard, we believe that our efforts were far from “minor”, and we are optimistic that this proof will serve as a valuable contribution to the subject of study.
>
> ---
>
> **SAM without normalization**
>
> > The theoretical analysis in Section 6.1/6.2 focus on a variant of SAM (SAM without normalization) for simplification, rather than the original SAM. While not a major issue, the formulation should be clearly presented in Section 6 to prevent potential reader confusion.
>
> We thank the reviewer for the suggestion and relate to the reviewer’s concern. While we have mentioned this already in Line 376 and Line 478, we will make this more clear by providing the explicit formulation of the un-normalized SAM in the main text.

---

> ### Author Response · Authors · 2024-11-14
> **Response to 18YL (2/2)**
>
> **Q: Adequate theoretical support for Section 3?**
>
> > Could the authors provide adequate theoretical support for the main findings? e.g., an analysis on diagonal linear networks.
>
> We sincerely appreciate the reviewer for providing constructive feedback. However, we find this task, even with simplified models, unmanageable. Specifically, it is not possible to increase the number of parameters in diagonal linear networks since they are entirely determined by the dimensionality of the data (i.e., input and output dimensions). Also, while we could attempt to make it somewhat possible for a similar variant of deep linear networks [5], it won’t affect the model capacity to fit more complex data and overfit, which is a crucial property when analyzing the generalization characteristics.
>
> Nonetheless, we can develop connections between Sections 3 and 6 in the following sense:
> - Section 6.1: As explained in **Section 6.1** above, connecting the observation in Section 4.2 and  Equation (6) suggests larger models can lead to flatter minima, which has been observed to correlate well with good generalization [6-9].
> - Section 6.2: We can also connect this with Equation (6), since unlike the under-parameterized scenario where SAM needs diminishing step-size [10], SAM can use larger constant step-sizes when the model is overparameterized, which also tightens the upper bound on the maximum Hessian eigenvalue and lead to flatter minima.
>
> Also, we remind that Section 4 directly supports the empirical phenomenon in Section 3. If there is any concern with the explanations in Section 4, please point us to it.
>
>
> ---
>
> **Closing remarks**
>
> We sincerely thank the reviewer for taking the time to review our work. While we hope that our response has addressed the reviewer’s concerns reasonably well should there remain any remaining concerns, please let us know so we can fix them further during the discussion period. Otherwise, given the reviewer’s positive assessment on soundness, presentation, and contribution (2/3/2), we find the overall rating score of 3 is quite discouraging, and hence, we sincerely hope that the reviewer could give it a reconsideration.
>
> &nbsp;
>
> **References** \
> [1] Ma et al. The power of interpolation: Understanding the effectiveness of SGD in modern over-parametrized learning. ICML, 2018. \
> [2] Liu and Belkin. Accelerating SGD with momentum for over-parameterized learning. ICLR, 2020. \
> [3] Meng et al. Fast and Furious Convergence: Stochastic Second-Order Methods under Interpolation. AISTATS, 2020. \
> [4] Vaswani et al. Painless Stochastic Gradient: Interpolation, Line-Search, and Convergence Rates. NeurIPS, 2019. \
> [5] Ji and Telgarsky, Gradient descent aligns the layers of deep linear networks, ICLR, 2019. \
> [6] Keskar et al. On large-batch training for deep learning: Generalization gap and sharp minima. ICLR, 2017. \
> [7] Chaudhari et al. Entropy-SGD: Biasing gradient descent into wide valleys. ICLR, 2017. \
> [8] Jiang et al. Fantastic generalization measures and where to find them. ICLR, 2020. \
> [9] Neyshabur et al. Exploring generalization in deep learning. NeurIPS, 2017. \
> [10] Andriushchenko and Flammarion. Towards understanding sharpness-aware minimization. ICML, 2022.

---

> ### Author Response · Authors · 2024-11-23
> **Dear Reviewer**
>
> Dear Reviewer 18YL
>
> We want to thank you again for your time and thoughtful feedback on our submission. To address your concerns, we have worked diligently to provide detailed clarifications and discussions in our rebuttal. Given that some time has passed since we shared our responses, we would greatly appreciate it if you could review our rebuttal, check whether our response addresses the concerns, and re-evaluate the score accordingly.
>
> We are more than happy to address any additional questions or comments you might have.
>
> Best wishes,\
> The authors.

---

> ### Comment · Reviewer_18YL · 2024-11-28
>
> Thank you for your detailed response. While I appreciate the effort invested in addressing my concerns, some key issues remain unresolved:
>
> - Section 6.1: Theorem 6.3 still does not sufficiently explain why overparameterization leads to flatter minima.
>   - Firstly, the comparison of results under different levels of overparameterization, with identical $\eta$ and $\rho$ in SAM, fails to demonstrate any specific benefits of overparameterization. The bound remains unchanged regardless of the degree of overparameterization. (By the way, I conjecture that greater overparameterization might still enhance generalization, even when using the same $\rho$.)
>   - Secondly, as noted in your response, greater overparameterization enables the use of a larger $\rho$, which contributes to flatter minima. However, the critical question remains: why does greater overparameterization permit a larger $\rho$? This causal relationship is not established by the theorem.
>
> - Section 6.2: While it is well-established that SGD converges linearly under overparameterization (Bassily et al., 2018), the linear convergence of SAM in this context is unsurprising. A more persuasive analysis would examine whether increased overparameterization accelerates convergence.
>
> Despite these key unresolved issues, I recognize the substantial effort behind this work. To encourage further research on these questions, I have revised my rating from 3 to 5.

---

> > ### Author Response · Authors · 2024-12-02
> > **Addressing additional response (1/2)**
> >
> > We greatly appreciate the reviewer’s recognition of our efforts. We have addressed specific points below.
> >
> > ---
> >
> > > why does greater overparameterization permit a larger $\rho$?
> >
> > We have presented theoretical and intuitive accounts of why greater overparameterization may have to permit larger $\rho$ already in Appendix D. We explain it further here in the following.
> >
> > The core premise is that SAM’s advantage over SGD hinges on the effectiveness of the perturbation step, which diminishes with greater overparameterization unless $\rho$ is adjusted. To ensure SAM retains its distinct benefits, $\rho$ must scale with the degree of overparameterization. We develop two accounts for this:
> >
> > First, if we consider the expected effect of perturbation $\epsilon \in\mathbb{R}^d$ of size $\rho$ on the individual parameter simply as $\mathbb{E}_i[\epsilon_i^2]=||\epsilon||_2^2/d=\rho^2/d$, we can see that $\mathbb{E}_i[\epsilon_i^2] \rightarrow 0$ as $d\rightarrow\infty$, which implies that **it would eventually have almost no effect on each parameter as the model scales unless $\rho$ is increased**.
> >
> > Also, assuming that the model gets smoother with overparameterization (which we empirically confirmed already in Figure 22-(a)), the Lipschitz bound on the gradients reveals that $\left|\left|\nabla f\left(x + \rho \frac{\nabla f(x)}{||\nabla f(x)||_2}\right) - \nabla f(x)\right|\right|_2 \leq \beta \left|\left|x + \rho \frac{\nabla f(x)}{||\nabla f(x)||_2} - x\right|\right|_2 = \beta \rho$, indicating that the SAM gradient becomes more similar to the original gradient as the model gets smoother (i.e., smaller smoothness constant $\beta$) with increasing size. This certainly indicates that **a larger perturbation bound is required to maintain similar levels of perturbation effect**.
> >
> > Much to our surprise, this aligns very well with our empirical observations in Figure 4: through an extensive hyperparameter search to identify the $\rho$ value that allows SAM to perform optimally (i.e. $\rho^\star$), we observed that the best-performing $\rho^\star$ consistently increases with overparameterization.
> >
> > All these results collectively point to the **necessity of increasing $\rho$ for SAM to fully exploit the advantages of overparameterization**.
> >
> > ---
> >
> > > By the way, I conjecture that greater overparameterization might still enhance generalization, even when using the same $\rho$.
> >
> > This is _NOT_ true. In fact, Figure 4 indicates that depending on $\rho$, overparameterization can impede the generalization improvement by SAM (i.e., SAM-SGD gap), or even cause SAM to underperform SGD.
> >
> > For instance, for CIFAR-10/ResNet-18, when $\rho = 0.005$ which is the optimal $\rho^\star$ for parameter count $d=45\text{k}$, increasing the parameter count to $d=701\text{k}$ leads to SAM underperforming SGD by -0.06%p. Even after scaling the model to $d=178.6\text{m}$, it still results in a decrease in the SAM-SGD gap.
> >
> > `Figure 4: CIFAR-10/ResNet-18`
> >
> > ||45k|176k|701k|2.8m|11.2m|44.7m|178.6m|
> > |-|-|-|-|-|-|-|-|
> > |$\rho=0.005$|**+0.36**|+0.44|-0.06|+0.27|+0.16|+0.48|+0.14|
> > |$\rho=0.1$|+0.04|**+0.66**|**+0.44**|**+0.84**|+0.82|+1.17|+0.90|
> > |$\rho=0.2$|-0.68|+0.07|+0.23|+0.80|**+0.96**|**+1.22**|**+1.01**|
> >
> > In contrast, a larger value of $\rho=0.2$ is highly effective in higher parameter counts, despite it degrading performance of SAM in $d=45\text{k}$.
> >
> > These results demonstrate the critical role $\rho$ in understanding how overparameterization achieves this level of effectiveness in enhancing SAM’s generalization.

---

> > > ### Author Response · Authors · 2024-12-02
> > > **Addressing additional response (2/2)**
> > >
> > > > the linear convergence of SAM in this context is unsurprising
> > >
> > > With respect, we believe there is a significant difference between anticipating a result based on intuition and rigorously proving it. Non-trivial work was put into deriving the linear convergence rate of SAM under overparameterization (e.g, development of Lemma J.1 and J.2). This, to our knowledge, is the first work to explicitly prove this result, which is much faster than the previously known sublinear rate.
> > >
> > > ---
> > >
> > > > A more persuasive analysis would examine whether increased overparameterization accelerates convergence.
> > >
> > > We definitely relate to the reviewer’s comment. However, as mentioned in Section 7, “Theoretical account of Section 3”, to the best of our knowledge, we are unaware of any theories or analyses that capture varying degrees of overparameterization applicable to our study. We conjecture that formulating such a notion in the context of interpolation is complex, and our understanding as a society is still limited. Nonetheless, we would be keen to find any such work, so please let us know if the reviewer could suggest any. We are eager to pursue further study in this direction.
> > >
> > > ---
> > >
> > > **Section 4 supports Section 3**
> > >
> > > While the reviewer raises a concern regarding Section 6 not supporting Section 3, as mentioned throughout our previous response (as well as in the paper), we are presenting **Section 4** to provide explanation of **Section 3**, not with Section 6. We would appreciate it if you could check out Section 4 and let us know whether there are any concerns regarding it.
> > >
> > > ---
> > >
> > > **Final remark**
> > >
> > > We sincerely thank the reviewer for taking the time to review our work. With all due respect, we are somewhat discouraged by Section 6 being judged solely from the perspective of developing an entirely new theory to fully explain Section 3. We believe our contributions in Section 6 lie in rigorously examining the influences of established theoretical notions of overparameterization on SAM while providing our best direct explanation for the observations in Section 4, none of which were explored in prior work. We hope the reviewer recognizes the contributions of these results in initiating and advancing toward a comprehensive understanding of the critical influence of overparameterization on SAM.

---

> > > > ### Author Response · Authors · 2024-12-04
> > > > **Closing remark**
> > > >
> > > > Once again, we express our sincere gratitude for the time and effort you have dedicated to reviewing our work and for your valuable feedback to further enhance our paper. We promise to incorporate suggestions that have yet to be included in the final manuscript. Also, we hope that we have adequately addressed most of your concerns. If so, we would greatly appreciate it if the reviewer could give a re-consideration to the rating of this work.

---

### Official Review · Reviewer_bQqz · 2024-11-04

**Soundness:** 3
**Presentation:** 4
**Contribution:** 3
**Rating:** 6
**Confidence:** 5

**Summary:**

The authors perform experiments to
measure the effect of overparameterization in SAM for a diverse set of tasks (Section 3). The goal is to observe how overparameterization
affects SAM under various conditions, e.g.,  label noise, sparsity, and regularization.
Furthermore, they prove that stable minima of SAM are flatter and have more uniform Hessian moments (if compared with SGD), and stochastic SAM can also converge at a linear. The overall contribution is that they empirically and theoretically proved that overparameterization critically affects SAM.

**Strengths:**

- interesting and well-motivated problem
 - very well written

**Weaknesses:**

- discussion on higher moments of Hessian is missing

**Questions:**

This is an interesting paper. I have a question: how does the convergence of higher-order moments of Hessian in your result compare with the other approaches in the literature, e.g., [1]? Can you provide a literature review on the previous works considering higher-order moments of Hessian to define flatness?

[1] Tahmasebi, Behrooz, et al. "A Universal Class of Sharpness-Aware Minimization Algorithms." Forty-first International Conference on Machine Learning.

---

> ### Author Response · Authors · 2024-11-24
> **Response to bQqz**
>
> We really appreciate the reviewer’s positive and constructive feedback. We are pleased that the reviewer finds our work to be interesting and well-motivated. While we respond to the reviewer’s specific comments as below, please do let us know if there is anything else we need to address further.
>
> ---
>
> **Higher-order moments of Hessian and its relation to flatness**
>
> We sincerely appreciate your interest and insightful suggestion regarding higher-order Hessian moments, and especially for bringing our attention to [1]. Upon investigation, we have identified several works relevant to this subject, which we detail below.
>
> First, [1] proposes a general framework for various new sharpness measures (determinant and Frobenius norm of Hessian, etc.) and their accompanying sharpness-aware optimizers. Here, we can define the higher-order power of Hessian as a measure of sharpness through the “homogeneous polynomial of degree $n$” described in Table 1 (fifth line), which, when combined with an expectation over mini-batches, can yield higher-order Hessian moments similar to those in our work. We believe this could shed new light on understanding the implications of bounding higher-order Hessian moments, thus very relevant to our work.
>
> In other works on higher-order Hessian moments and their relationship with sharpness, the linear stability analysis for SGD in [2] has shown that it requires additional condition on the second-order Hessian moment (non-uniformity) compared to GD, which has been empirically observed to be highly correlated with sharpness. Building on this, [3] suggests that in higher-order linearly stable minima of SGD, the higher-order gradient moments are also bounded, which is then shown to be a contributing factor for improved generalization performance of SGD. On the contrary, SAM possesses bounds for higher-order Hessian moments without assuming higher-order linear stability, which might partially explain its better generalization performance.
>
> It is worth noting, however, that we are not proposing higher-order moments of the Hessian as a new notion of flatness unlike [1]. Instead, we highlight that the existence of an upper bound for higher-order Hessian moments in the linearly stable minima found by SAM is noteworthy. This property of SAM, to the best of our knowledge, has not been reported in prior studies.
>
> Nonetheless, we once again extend our gratitude to the reviewer for suggesting [1], which we will include in the revised version along with these discussions.
>
> &nbsp;
>
> **References** \
> [1] Tahmasebi et al. "A Universal Class of Sharpness-Aware Minimization Algorithms." ICML, 2024. \
> [2] Wu et al., How SGD Selects the Global Minima in Over-parameterized Learning: A Dynamical Stability Perspective. NeurIPS, 2018. \
> [3] Ma and Ying., On Linear Stability of SGD and Input-Smoothness of Neural Networks. NeurIPS, 2021.

---

> > ### Comment · Reviewer_bQqz · 2024-11-25
> >
> > I sincerely thank the authors for comprehensively responding to my comments/concerns, and I assure them that I read their rebuttal in detail. For now, I don't have any new particular questions/concerns, and I will decide whether to keep my score or increase it after a bit of necessary discussion with other reviewers/AC. Thanks!

---

> > > ### Author Response · Authors · 2024-11-25
> > > **Thank you**
> > >
> > > We sincerely thank the reviewer for the thoughtful consideration of our rebuttal. We will ensure that these new discussions are reflected in the revised version, and remain available to address any further concerns that may arise during the remainder of the discussion period.

---

### Meta-Review · Area_Chair_5Yj1 · 2024-12-05

**Metareview:**

This paper examines the impact of overparameterization on sharpness-aware minimization, focusing on factors such as label noise, sparsity, and regularization. The reviewers found the topic to be well-motivated and the experiments extensive. However, major concerns were raised regarding the empirical and theoretical grounding of key claims, which were at times perceived as overstated or predictable. Despite the significant discussions the paper has generated, the initial evaluations remained lukewarm. The authors are encouraged to incorporate the important feedback given by the knowledgeable reviewers.

**Additional Comments On Reviewer Discussion:**

There was no particular excitement due to the issues outlined above, which remained largely unresolved despite the discussions.

---

### Decision · Program_Chairs · 2025-01-22

Reject